# Neoadjuvant tislelizumab plus stereotactic body radiotherapy and adjuvant tislelizumab in early-stage resectable hepatocellular carcinoma: the Notable-HCC phase 1b trial

Zhongchao Li [1,6], Jing Liu[2,6], Bo Zhang [1,6], Jinbo Yue[2,6], Xuetao Shi [1], Kai Cui [1], Zhaogang Liu[1], Zhibin Chang[1,3], Zhicheng Sun [1,3], Mingming Li [1,3], Yue Yang [1,3], Zhao Ma[4], Lei Li[1], Chengsheng Zhang[1], Pengfei Sun [1], Jingtao Zhong [1] & Lei Zhao [1,3,5] ✉

Notable-HCC (NCT05185531) is a phase 1b trial, aiming to evaluate the safety and preliminary effectiveness of neoadjuvant PD-1 blockade plus stereotactic body radiotherapy (SBRT) in early-stage resectable hepatocellular carcinoma (HCC). Twenty patients with HCC of BCLC stage 0-A received 3× Gy SBRT and two cycles of tislelizumab, an anti-PD-1 monoclonal antibody before the curative HCC resection. Primary endpoints were the surgery delay, radiographic and pathological tumor response after the neoadjuvant therapy, safety and tolerability. During the neoadjuvant therapy, treatment-related adverse events (TRAEs) of grade 1-2 occurred in all 20 patients (100%), eight patients (40%) had grade 3 TRAEs, no grade 4 to 5 TRAE occurred, and all resolved without corticosteroids treatment. Per mRECIST, the objective response rate was 63.2% (12/19), with 3 complete response; the disease control rate was 100%. Two (10.5%) patients achieved complete pathological response. No surgery delay occurred. The neoadjuvant therapy did not increase the surgical difficulty or the incidence of complications. Secondary endpoints of disease-free survival and overall survival were not mature at the time of the analysis. Our pilot trial shows that neoadjuvant therapy with anti-PD-1 + SBRT is safe and promotes tumor responses in early-stage resectable HCC.

Primary liver cancer (PLC) is the third leading cause of cancer-related mortality worldwide[1]; in China, it remains the second most lethal cancer type for both males and females between 2005 to 2020, and in 2020, it has become the top leading cause of cancer-related death in population aged 20–39 and 40–59 years[2]. Hepatocellular carcinoma (HCC) represents ~75–85% of PLCs[1]. Surgical resection remains the mainstay curative-intent treatment for early-stage HCC, whilst unfortunately, only 10–30% of HCC patients are candidates for liver-directed therapy[3]; in our center, only <20% of HCC patients were resectable upon diagnosis[4]. Even after radical resection, the 5-year recurrence

[1]Department of Hepatobiliary Surgery, Shandong Cancer Hospital Affiliated to Shandong First Medical University, 440 Jiyan Road, Huaiyin District, Jinan, China. [2]Department of Abdominal Radiation Oncology, Shandong Cancer Hospital Affiliated to Shandong First Medical University, 440 Jiyan Road, Huaiyin District, Jinan, China. [3]Shandong First Medical University and Shandong Academy of Medical Sciences, 6699 Qingdao Road, Huaiyin District, Jinan, China. [4]The Fourth People's Hospital of Jinan, Jinan, China. [5]The Affiliated Cancer Hospital of Xinjiang Medical University, Urumqi, China. [6]These authors contributed equally: Zhongchao Li, Jing Liu, Bo Zhang, Jinbo Yue. ✉e-mail: drzhaolei@hotmail.com

rate of HCC is up to 70%, posing a big challenge for the long-term survival of patients[3,5,6]. The pattern of recurrence is typically within the liver, which might originate from residual micro-metastatic disease or de novo tumorigenesis following resection[7,8], highlighting the potential benefit of perioperative antitumor therapy to reduce recurrence and improve long-term outcomes. Adjuvant immune checkpoint inhibitor-based systemic therapy was recently recommended by the AASLD guideline for the treatment of HCC in patients at high risk of recurrence after liver resection[9]. Currently, no neoadjuvant therapies are recommended for early-stage HCC by the primary HCC guidelines (e.g., NCCN, EASL, AASLD).

The results from Himalaya trial[10] and IMbrave 150 trial[11] firmly established the pivotal roles of immune checkpoint inhibitors (ICIs) in the systemic treatment of HCC. Neoadjuvant therapy strategies incorporating anti-PD-(L)1 antibodies, alone or in combination, have shown encouraging radiographic and pathological responses in multiple tumor types[12,13]. Similar explorations have been done in early-stage HCC. In a randomized, open-label phase II trial of perioperative nivolumab alone or nivolumab plus ipilimumab in resectable HCC, Kaseb and colleagues reported a major pathological response (MPR) rate of 27% in 11 patients who underwent surgical resection after neoadjuvant nivolumab plus ipilimumab, and patients who had MPR were associated with a longer recurrence-free survival (RFS) compared with those without MPR[14]. More recently, investigators of a single-arm, open-label, phase II trial reported a significant tumor necrosis (defined as more than 70% necrosis of resected tumor) rate of 20% and an objective response rate (ORR) of 15% in 20 patients treated by two cycles of neoadjuvant cemiplimab[15]. These findings suggest the feasibility and efficacy of neoadjuvant immunotherapy for resectable HCC, and support the explorations for novel combination strategies to further strengthen the clinical benefits in this setting.

Radiotherapy has emerged as a promising locoregional treatment option for HCC[16,17]. A randomized control trial (RCT) showed that, for patients with a single and small HBV-related HCC at high risks of microvascular invasion, neoadjuvant intensity-modulated radiotherapy (IMRT) provided a promising response rate with mild toxicity[18]. Mounting evidence shows the synergistic effects between radiotherapy and immunotherapy[19–24]. Radiotherapy might be a potent immunomodulator, enhancing the efficacy of ICIs through multiple mechanisms, including induction of immunogenic cell death with release of neoantigens, upregulation of major histocompatibility complex (MHC) and enhanced antigen presentation, activation of dendritic cells, and enhanced antigen cross-presentation, modulation of checkpoint expression, and increased T cell infiltration into the tumor[25]. In a recent single-center, randomized phase II trial in early-stage non-small cell lung cancer, Altorki and colleagues reported that the combination of stereotactic body radiotherapy (SBRT) with neoadjuvant durvalumab was well tolerated and associated with a significantly high MPR rate of 53.3% (16/30) against 6.7% (2/30) with durvalumab monotherapy, suggesting that the combination of SBRT with PD-(L)1 inhibitor in a neoadjuvant setting may enhance the effect of immunotherapy and improve the antitumor effecacy[22]. Similar pilot early-stage trials are still very limited, but have been reported in head and neck squamous cell carcinomas (HNSCC)[26], triple-negative breast cancer (TNBC)[27], and esophageal squamous cell cancer[28]. Additionally, a case series with five uHCC patients receiving SBRT plus anti-PD-1 antibodies revealed encouraging response, with all patients achieving a partial response (PR) per Response Evaluation Criteria in Solid Tumors (REICIST) criteria, and when assessed by modified RECIST (mRECIST) criteria, two patients achieving complete response (CR), and three having PR[29].

In this work, we hypothesized that SBRT could be safely administered with a PD-(L)1 inhibitor, and may augment immunological and clinical response in patients with early-stage resectable HCC. Tislelizumab is an anti-PD-1 antibody that, in the phase III randomized

RATIONALE-301trial, demonstrated noninferior overall survival (OS) benefit to sorafenib[30], and has been approved in China as both the first-line and the second-line treatment for patients with unresectable HCC. Here we report the safety and efficacy of SBRT plus tislelizumab as neoadjuvant therapy in early-stage HCC. Neoadjuvant therapy with SBRT combined with tislelizumab appears to be safe and well-tolerated, promoting tumor responses as well as antitumor immunity. Our pilot results warrant further studies of neoadjuvant therapy with ICI plus radiotherapy in resectable HCC.

## Results
### Patient characteristics
Between 22 March 2022 and 17 July 2023, 22 patients were screened for eligibility; one was excluded due to inadequate liver function for surgical resection, and one was not suitable for SBRT treatment (supplementary fig. 2). Finally, a total of 20 patients were enrolled, with a median age of 58.5 years (range, 48–78). Nineteen patients had an ECOG PS of 0 and one of 1, and all 20 patients had a Child-Pugh score of A5 at the baseline. HCC were newly diagnosed in 16 patients, and 4 patients had recurrent HCC, the time intervals between their first HCC resection to the recruitment were: pts no. 2, 49 months; pts no.3, 13 months; pts no.11, 95 months; pts no.13, 22 months. The most common underlying cause of HCC was HBV infection (n = 17, 85.0%). Three (15.0%) were at Barcelona Clinic Liver Cancer (BCLC) stage 0, and other 17 patients were all of BCLC stage A; while according to the China Liver Cancer Staging (CNLC) system, seventeen (85.0%) patients were at stage Ia and 3 (15%) at stage Ib (supplementary table 1).

### Safety profile of neoadjuvant SBRT+ anti-PD-1
During the neoadjuvant therapy period, treatment-related AEs (TRAEs) of any grade occurred in all 20 patients (Table 1). The three most frequent TRAEs were decreased lymphocyte count (n = 18, 90.0%), decreased platelet count (n = 14, 70.0%), and decreased white blood cell count (n = 13, 65.0%). Eight (40%) patients experienced grade 3 TRAEs, and no grade 4 to 5 TRAE occurred. The most common (≥15%) grade 3 TRAEs were decreased lymphocyte count (n = 3, 15.0%) and decreased neutrophil count (n = 3, 15.0%). Serious adverse events (SAE) occurred in one (5.0%) patient (no. 12) due to ALT/AST increase after first cycle of tislelizumab, presenting with mild fatigue, and the second cycle of tislelizumab was cancelled. The level of ALT/AST of this patient dropped to normal, and thus the patient received the curative HCC resection on day 56. All adverse events resolved spontaneously without the need for corticosteroids treatment.

In 8 patients with grade 3 TRAEs, 6 of them had myelosuppression, manifested as decreases of lymphocyte, neutrophil, WBC and platelet. We further explored the baseline and dynamic changes of the WBC counts in these 6 patients during the neoadjuvant therapy. When compared with the baseline levels, no deteriorations of laboratory abnormalities with shifts of ≥3 toxicity grade levels occurred during the neoadjuvant therapy (supplementary table 2, supplementary table 3).

### Surgeries after neoadjuvant SBRT+ anti-PD-1
None of patients (0%) postponed surgery over 6 weeks after neoadjuvant therapy due to any causes (Fig. 1). All 20 patients (100%) underwent laparotomic operation, with one (5%) undergoing radiofrequency ablation, while the remaining 19 patients (95%) achieved curative R0 resection. The median time from initiation of neoadjuvant therapy to surgical resection was 55.9 days (range 49–69); surgeries delay over 10 days occurred in four patients, with three attributed to COVID-19 infection (pts no.5, 19 days; pts no.15, 10 days; pts no.19, 11 days), and one (pts no. 8, 19 days) due to poor liver function, despite compromised liver function at enrollment. Compared to regular HCC resection, the neoadjuvant therapy of tislelizumab plus SBRT did not

**Table 1 | Treatment-related adverse events during neoadjuvant treatment**

| | Any grade | Grade 3 |
|---|---|---|
| Any TRAE | 20 (100.0%) | 8 (40.0%) |
| Lymphocyte count decrease | 18 (90.0%) | 3 (15.0%) |
| Platelet count decrease | 14 (70.0%) | 1 (5.0%) |
| White blood cell decrease | 13 (65.0%) | 2 (10.0%) |
| Aspartate aminotransferase increase | 9 (45.0%) | 2 (10.0%) |
| Hypoalbuminemia | 9 (45.0%) | 0 |
| Neutrophil count decrease | 8 (40.0%) | 3 (15.0%) |
| Anemia | 7 (35.0%) | 0 |
| Alanine aminotransferase increase | 5 (25.0%) | 1 (5.0%) |
| Blood lactate dehydrogenase increase | 5 (25.0%) | 0 |
| Abdominal distension | 5 (25.0%) | 0 |
| Fatigue | 5 (25.0%) | 0 |
| Nausea | 5 (25.0%) | 0 |
| Electrocardiogram T wave abnormal | 5 (25.0%) | 0 |
| Blood bilirubin increase | 4 (20.0%) | 0 |
| Alkaline phosphatase increase | 4 (20.0%) | 0 |
| Hyponatremia | 4 (20.0%) | 0 |
| GGT increase | 3 (15.0%) | 0 |
| Fibrinogen decrease | 3 (15.0%) | 0 |
| Hypokalemia | 2 (10.0%) | 0 |
| Hypothyroidism | 2 (10.0%) | 0 |
| Activated partial thromboplastin time prolonged | 2 (10.0%) | 0 |
| Ascites | 1 (5.0%) | 0 |
| Cardiac troponin T increase | 1 (5.0%) | 0 |
| INR increase | 1 (5.0%) | 0 |
| Thyroid stimulating hormone increase | 1 (5.0%) | 0 |
| Abdominal pain | 1 (5.0%) | 0 |
| Pneumonitis | 0 | 0 |

Data are n (%). *TRAE* treatment-related adverse event.

increase the surgical difficulties and the risk of postoperative complications (supplementary results, supplementary table 4).

### Radiographic and pathological tumor responses

Radiographic and pathological responses after neoadjuvant treatment are shown in Fig. 2. Radiological response was evaluated in 19 patients in the EAS. After the neoadjuvant therapy, per RECIST v1.1, 42.1% (8/19) patients had an objective response, all were PR, no patients achieved CR, and the other 11 (57.9%) patients were SD, so the DCR was 100%. While per mRECIST, 63.2% (12/19) patients had an objective response, with 3 achieving CR and 9 achieving PR, the DCR was also 100%. All 19 patients received curative R0 HCC resection successfully; two (10.5%) out of the 19 patients achieved pCR, and 6 (31.6%) reached MPR+pCR (supplementary table 5).

### Adjuvant therapy

Fifteen of 19 patients who underwent curative HCC resection received regular adjuvant therapy of tislelizumab per protocol. Four patients did not receive regular adjuvant therapy, and the reasons were: pts no. 2, discontinued after 4 cycles due to cerebral infarction; pts no. 6, discontinued after 3 cycles due to covid-19 pneumonia; pts no. 8 chose active surveillance due to poor liver function, and pts no. 12 chose active surveillance due to SAE of increased ALT/AST during the neoadjuvant therapy.

During the adjuvant therapy, TRAEs of any grade occurred in all 17 patients who received adjuvant tislelizumab after the R0 HCC resection, with the vast majority being grade 1 to 2. The three most common

types TRAEs were decreased lymphocyte count ($n = 13$, 76.5%), decreased platelet count ($n = 12$, 70.6%), and decreased white blood cell ($n = 7$, 41.2%). Three patients experienced grade 3 TRAEs, which were decrease lymphocyte count ($n = 3$, 17.6%) and decreased platelet count ($n = 1$, 5.9%) (supplementary table 6). All TRAEs during adjuvant therapy were asymptomatic, and no corticosteroids treatment was needed. There was no grade 4 or 5 TRAE occurred.

Again, we further explored the baseline (upon being recruited and before the initiation of the neoadjuvant therapy) and dynamic changes of the WBC counts in these 3 patients with grade 3 TRAEs during the adjuvant therapy. When compared to the baseline levels and the levels during the period of adjuvant therapy, shifts of $\geq 3$ toxicity grade levels (from grade 0 to 3) occurred in 2 patients (decreased lymphocyte count) during the adjuvant therapy (supplementary table 7, supplementary table 8).

Median time from enrollment to data cutoff (Dec 1st, 2023) for the current analysis was 6.1 months (IQR 5–12.5). The median cycle of adjuvant tislelizumab in 15 patients was 4 (range, 2–16). With a median follow-up of 4.0 months since the resection (range, 2.2–18.8), disease recurrence developed in only 1 out of 19 patients (5.3%, pts no. 8) (Fig. 3).

Representative MRI/CT images at the baseline, after the neoadjuvant therapy (normally one week before the surgery) and at the first follow-up (one month after the surgery), as well as the photos of the resected tumor specimen of all patients, are available in the Source data file.

### Biomarker analysis

As part of the exploratory analyses, the biomarker analyses of the antitumor immunity were predefined in the study protocol. We conducted ssGSEA using RNA-seq data obtained from patients' tumors pre- and post-neoadjuvant therapy of tislelizumab plus SBRT, revealing substantial alterations in immune infiltration patterns following the neoadjuvant therapy compared to the pre-therapy state. Notably, both adaptive and innate immune-related cell populations exhibited a significant overall increase in ssGSEA scores after the neoadjuvant therapy (Fig. 4). Specifically, within the spectrum of T cell-related subtypes, a significant augmentation ($P < 0.05$) was observed post-treatment across most cell types. Noteworthy enhancements were recorded in the ssGSEA scores of several subsets, including activated CD8 T cell, central memory CD4 T cell, central memory CD8 T cell, effector memory CD8 T cell, gamma delta T cell, regulatory T cell, T follicular helper cell (Tfh), and type 1 T helper cell. Moreover, a conspicuous elevation in the ssGSEA scores of innate immune-related cells, particularly dendritic cells, was noted after the neoadjuvant therapy.

In RNA-seq analyses of RNA from paired pre- and post-treatment specimens, a substantial upregulation of T-cell activation-related gene expression was observed following the treatment. This increase was clearly visualized in a heatmap with red areas indicating the higher gene expression levels (Fig. 5a). Concurrently, a significant increase in the corresponding immune scores was observed ($P < 0.01$, Fig. 5b). Furthermore, among patients with varying mRECIST responses, a distinct trend emerged in the post-treatment T cell activation-related gene immune scores, demonstrating a hierarchy of CR > PR > SD (Fig. 5c). Notably, patients achieving CR exhibited markedly higher T cell activation immune scores post-treatment compared to those PR patients, demonstrated by a trend toward significance ($P = 0.06$, Fig. 5c).

Furthermore, published gene signatures associated with tumor-reactive T cells, cytotoxic cells, naive T cells, monocyte-derived macrophages, and B cells exhibited consistent upregulation, marked by discernible red areas post-therapy in these patients (Fig. 6).

Substantial elevations in immune scores were specifically observed in signatures of tumor-reactive T cells ($P < 0.05$) and cytotoxic cells ($P = 0.07$) (Fig. 7a). Noteworthy is the significantly higher immune score observed in patients achieving CR compared to those

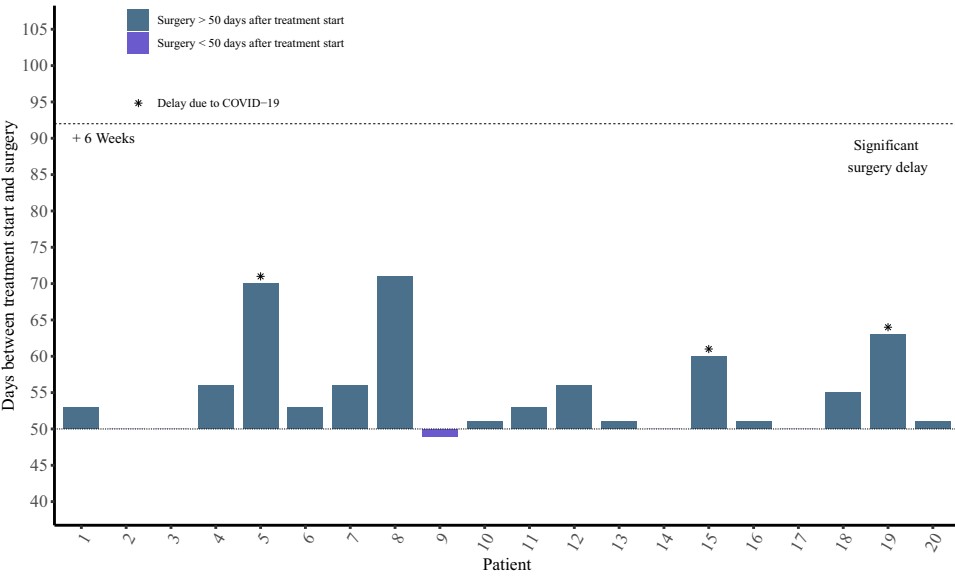

**Fig. 1 | Days between treatment initiation and surgery (*n* = 20).** Each bar represents one patient in safety analysis set. Source data are provided as a Source Data file.

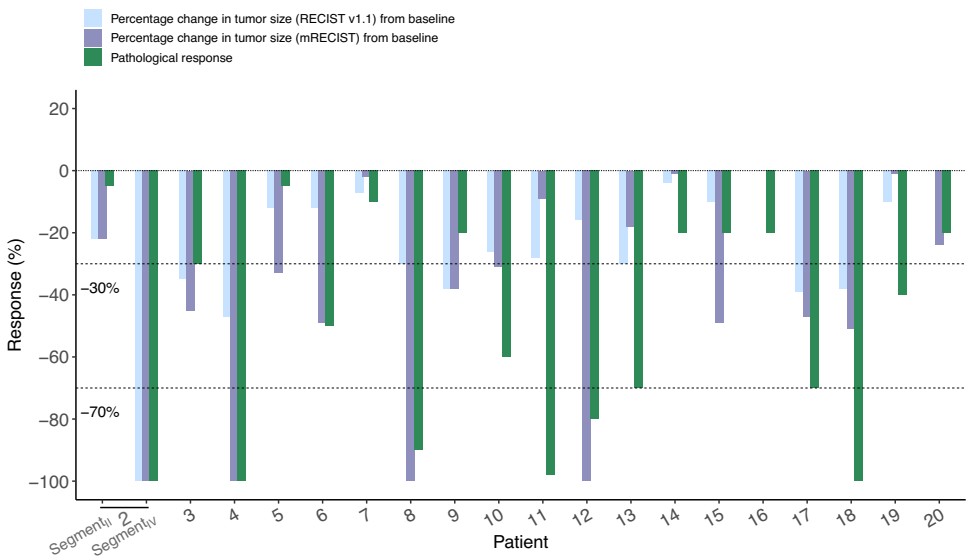

**Fig. 2 | Radiographic and pathological responses to neoadjuvant SBRT and tislelizumab in efficacy analysis set (*n* = 19).** Waterfall plots of the proportion of patients with an overall response by RECIST v1.1 and mRECIST before surgery, and major pathological responses on resected tumors. Source data are provided as a Source Data file.

with PR (*P* < 0.05) and SD (*P* = 0.07) in both tumor-reactive T cells and cytotoxic cells (Fig. 7b).

We further investigated the expression changes of HLA genes pre- and post- the neoadjuvant therapy. The analysis of the gene expression heatmaps revealed a significant upregulation in both HLA class I and class II genes following treatment (Fig. 8a). Notably, a more pronounced upregulation of HLA gene expression was evident in CR patients (Fig. 8a). Subsequent statistical analysis indicated a significant increase in immune scores after the neoadjuvant therapy (Fig. 8a). Both HLA class I and class II genes exhibited significantly higher immune scores across different mRECIST response levels, consistently demonstrating a trend of CR > PR > SD (Fig. 8c). Moreover, the differential expression of HLA class II genes appeared more substantial, particularly noteworthy in CR patients, significantly higher than SD patients (*P* < 0.05) (Fig. 8c).

To evaluate the similarity between TRB sequence repertoires pre- and post-neoadjuvant therapy, we compared the overlapping and newly generated CDR3 amino acid (aa) sequences in each patient post-

therapy in contrast to pre-therapy. Our findings revealed a notably larger proportion of newly generated clonotypes after the neoadjuvant therapy (Fig. 9a). Further analysis of clonotypes frequency distributions categorized all clonotypes into four groups based on their frequency: hyperexpanded, large, small, and rare clonotypes. Within the overlapping clonotypes, a significant rise in hyperexpanded clonotypes was evident among patients. More than 70% of clonotypes belonged to large and hyperexpanded groups, whereas newly generated clonotypes were predominantly large group, yet with fewer hyperexpanded clonotypes (Fig. 9b).

## Discussion

During the literature review, we found no report of trials exploring the safety and efficacy of the combination of radiotherapy with ICI at the neoadjuvant setting for early-stage resectable HCC patients. Other trials investigating mono-radiotherapy (NCT04587739, NCT05598060) or radiotherapy combined with/versus other treatments

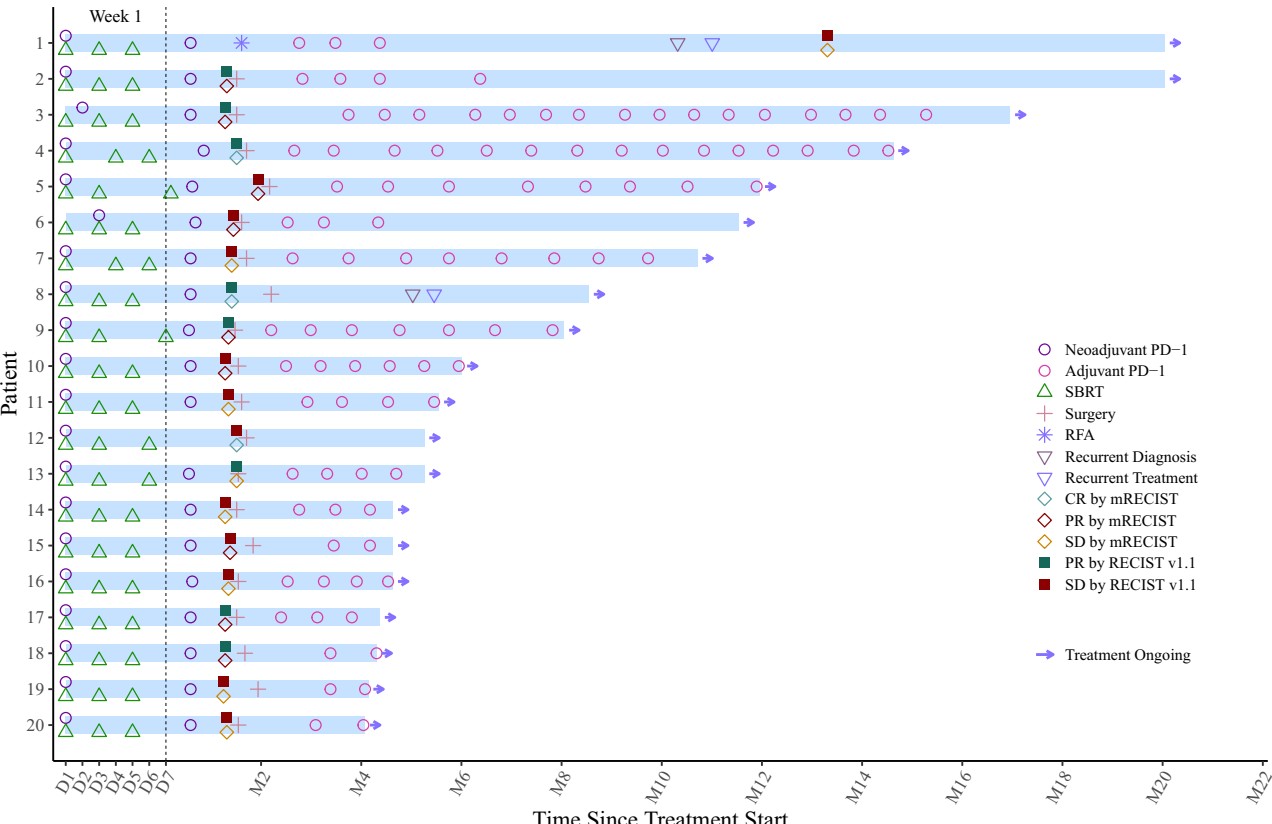

**Fig. 3 | The swimmer plot presents the treatment duration for patients who received neoadjuvant therapy, and subsequent adjuvant therapy after surgery.** Each bar represents one patient in the study ($n = 19$). Source data are provided as a Source Data file.

(NCT04857684, SBRT + Atezolizumab/Bevacizumab; NCT05137899, Atezolizumab/ Bevacizumab vs SBRT) are still ongoing.

In the treatment of malignant solid tumors, both adjuvant and neoadjuvant therapies play crucial roles in reducing the risk of recurrence and metastasis following curative resection, ultimately leading to improved survival outcomes. Benefiting from the recent advancements in tumor treatment, neoadjuvant therapy has been gradually recommended by major guidelines for certain types of malignant tumors[31]. HCC is highly invasive, even for patients with early-stage tumors at BCLC 0-A stage (corresponding to CNLC Ia-Ib stage), over one quarter (26.8%) of them recurred within the first year after the curative resection, and their 3-year and 5-year disease-free survival (DFS) was only 54.6% and 45.4%, respectively[32]. So, effective neoadjuvant therapy is an unmet need for HCC, including early-stage HCC. Also, for early-stage HCC, the better liver function reserve, good performance status, greater tolerance to potential immune-related toxicities, and lower risk of progression to unresectable tumor in case of being refractory to the treatment, provide additional rational for the neoadjuvant use of immunotherapy. This study aimed to explore the "neoadjuvant" therapy of HCC, which means the tumors must be clearly and definitely "resectable"; in BCLC staging system, resection/(ablation/transplantation) is only recommended to stage 0-A tumors as the first option; for tumors of stage B or beyond, the first option is not resection, thus the nature of the treatment would rather be "conversion" or "down-staging", instead of "neoadjuvant".

There are potential advantages of neoadjuvant therapy, including the shrinkage of tumor to limit the extent of curative surgery, allowing the pathological evaluation of the surgical specimens after the treatment, providing rapid and individualized evaluations of the treatment by tumor responses, etc[33].

On the other hand, there are also potential disadvantages of neoadjuvant therapy. Severe AE of neoadjuvant therapy can delay the resection, or increase the risk of postoperative morbidity, etc[33]. In this study, the neoadjuvant therapy of 3 fractions of SBRT (8 Gy) plus two cycles of tislelizumab (200 mg) was generally safe and well-tolerated, no grade 4 or 5 TRAE occurred. Grade 3 TRAEs were recorded in eight (40%) patients, which were mainly myelosuppression. In CTCAE v 5.0, the determination of the grade of blood/bone marrow-related AEs does not take their baseline levels into consideration, while for patients with a background of chronic liver diseases, myelosuppression is very common. In 20 participants, 19 had chronic hepatitis infection, over 25% showed abnormal baseline laboratory tests in WBC, lymphocyte, neutrophil, and platelet counts, and no deteriorations of ≥ 3 toxicity grade levels occurred.

WBC and platelet decrease, as well as ALT/AST increase, are very common even in mono-radiotherapy of liver cancer[34], since the radiotherapy of hepatic tumors will inevitably involve part of normal liver parenchyma into irradiated area, and lead to radiation-induced liver injury. Considering that the AE spectrums of radiotherapy and ICIs overlap with each other, it is technically difficult to definitely attribute these AEs to either one of them. Just like the enhanced antitumor effect, the AEs can be the result of their synergetic combination, too.

The major concern of neoadjuvant therapy is that the disease of non-responders may progress or even metastasize, and compromise their opportunity of curative surgery[33]. Therefore, it is crucial to explore combination strategies that can achieve rapid and high disease control rates in the neoadjuvant setting. In a phase II study evaluating neoadjuvant nivolumab with or without ipilimumab as neoadjuvant treatment for HCC[14], two patients in each group were unable to undergo surgery resection due to disease progression. In another single-arm phase II trial[15] of 20 HCC patients treated with two cycles of neoadjuvant cemiplimab, resection was aborted in one patient due to portal lymph node metastasis discovered during the surgical exploration.

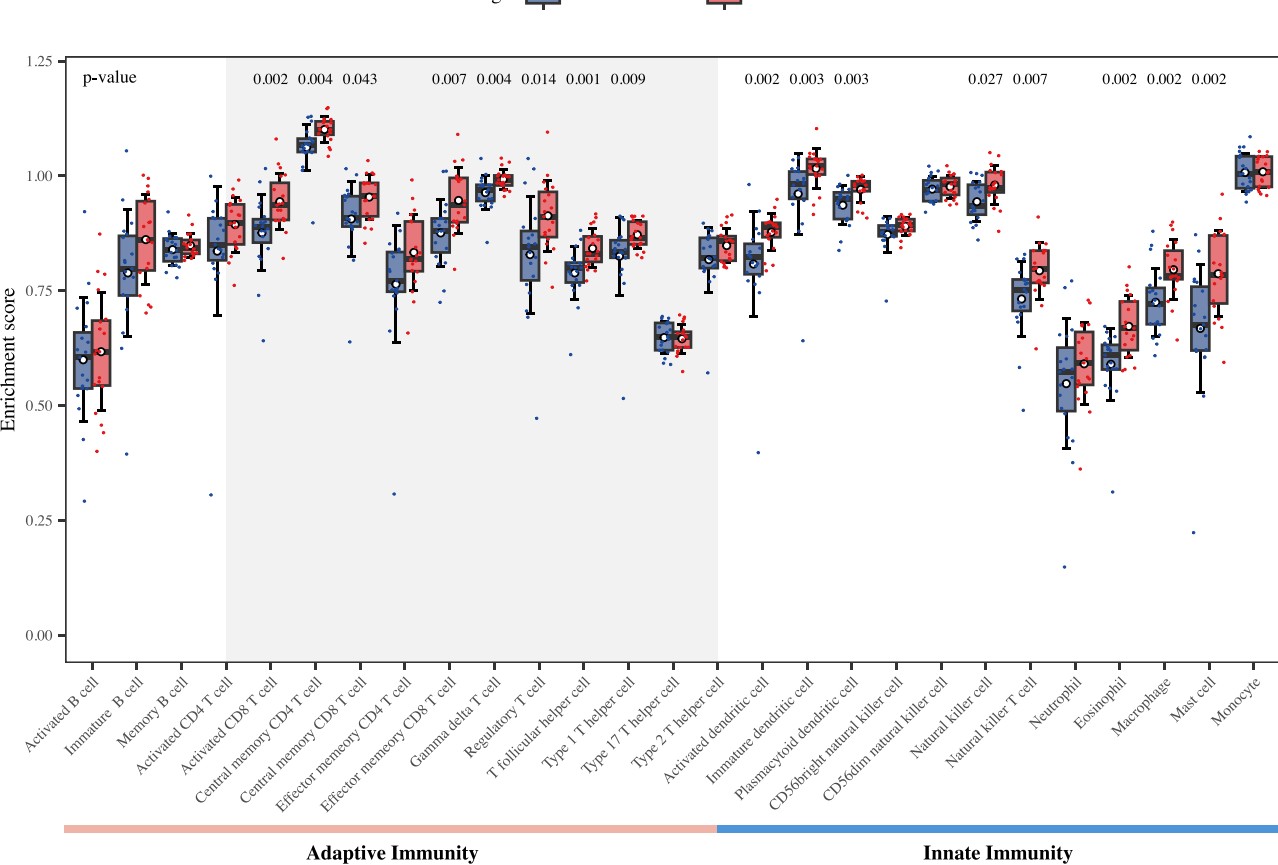

**Fig. 4 | Immune infiltration differential analysis of bulk RNA sequencing data between pre- and post-neoadjuvant therapy of tislelizumab plus SBRT in HCC patients (n = 19).** Blue represents patients at pre-neoadjuvant therapy (n = 19), and red represents patients at post-neoadjuvant therapy (n = 19). Points represent the enrichment scores estimated by ssGSEA for immune cells in each patient. Boxplot with error bar shows the distribution of enrichment scores of patients at pre- neoadjuvant therapy and post-neoadjuvant therapy for each immune cell type. Within each box, the horizontal line and box indicate the median and 25th/75th percentile, respectively. The white dot represents the mean, and error bar represents the mean ± standard deviation. The two-sided p values are performed by the Wilcoxon rank-sum test, and exact p value <0.05 are labeled above each comparison. Source data are provided as a Source Data file.

Notably, in our study, the combination neoadjuvant therapy of tislelizumab and SBRT resulted in clinically significant disease control, and all patients had radiographic tumor regression in varying degree.

All 19 patients who underwent hepatectomy reached R0 resection. A clinically meaningful major and complete pathological response was also observed after this combination neoadjuvant therapy. Our results presented a promising antitumor effect of anti-PD-1 plus SBRT, suggesting that immunotherapy combined with radiotherapy could be a potentially feasible clinical strategy for early-stage resectable HCC, as evidenced by the numerically higher ORR and high disease control rate observed in our study. From a more general point of view, our preliminary results could bring more follow-up large-scale trials to validate, whether the combination of a local-regional therapy to ICIs can significantly improve the local control rate, thus reducing the risk of cancellation of curative resection due to tumor progression after neoadjuvant therapy.

However, it is important to interpret these results cautiously due to several factors, including limited sample size, relatively earlier tumor stages, variations in study populations, and the indirect comparison between different trials. Further studies with larger sample sizes, incorporating a control group such as immune mono-therapy, are warranted to validate the clinical benefits of this combination therapy.

In our study, no patients had surgical cancellation or significant delay due to TRAEs or tumor progression. In addition, the combination neoadjuvant therapy did not increase the difficulty of surgical resection or the incidence of postoperative complications.

During the adjuvant therapy, the types of occurred TRAEs were very similar to the neoadjuvant phase, with the top three most frequently observed types being same. But the severities were significantly milder. The adjuvant therapy was mono- tislelizumab without radiotherapy; compared to the adjuvant therapy phase, the relieved severity and frequency of TRAEs indicated that TRAEs in the neoadjuvant phase may be the results of synergistic effects from ICIs and radiotherapy.

Recurrence of HCC was diagnosed in one patient (pts no. 8) three months after the resection, the patient received radiofrequency ablation treatment, and remained in tumor-free survival again thereafter. With the relatively short follow-up time, data for DFS and OS have not sufficiently matured and will be reported later.

When ICIs are combined with the surgical treatment of solid tumors, it is hypothesized that neoadjuvant ICI therapy may be more effective compared to adjuvant therapy. The presence of tumor antigens presented in the tumor before resection may prompt a stronger and more prolonged antitumor T cell immune response, allowing a more active efficacy against micro-metastatic foci compared to the adjuvant ICI approach. Neoadjuvant ICIs, both in animal models[35] and in human patients[36], had shown that T cell augmentation is indeed more significant when ICI is given before the removal of the tumor compared to when given after the resection. In the clinical treatment of stage III

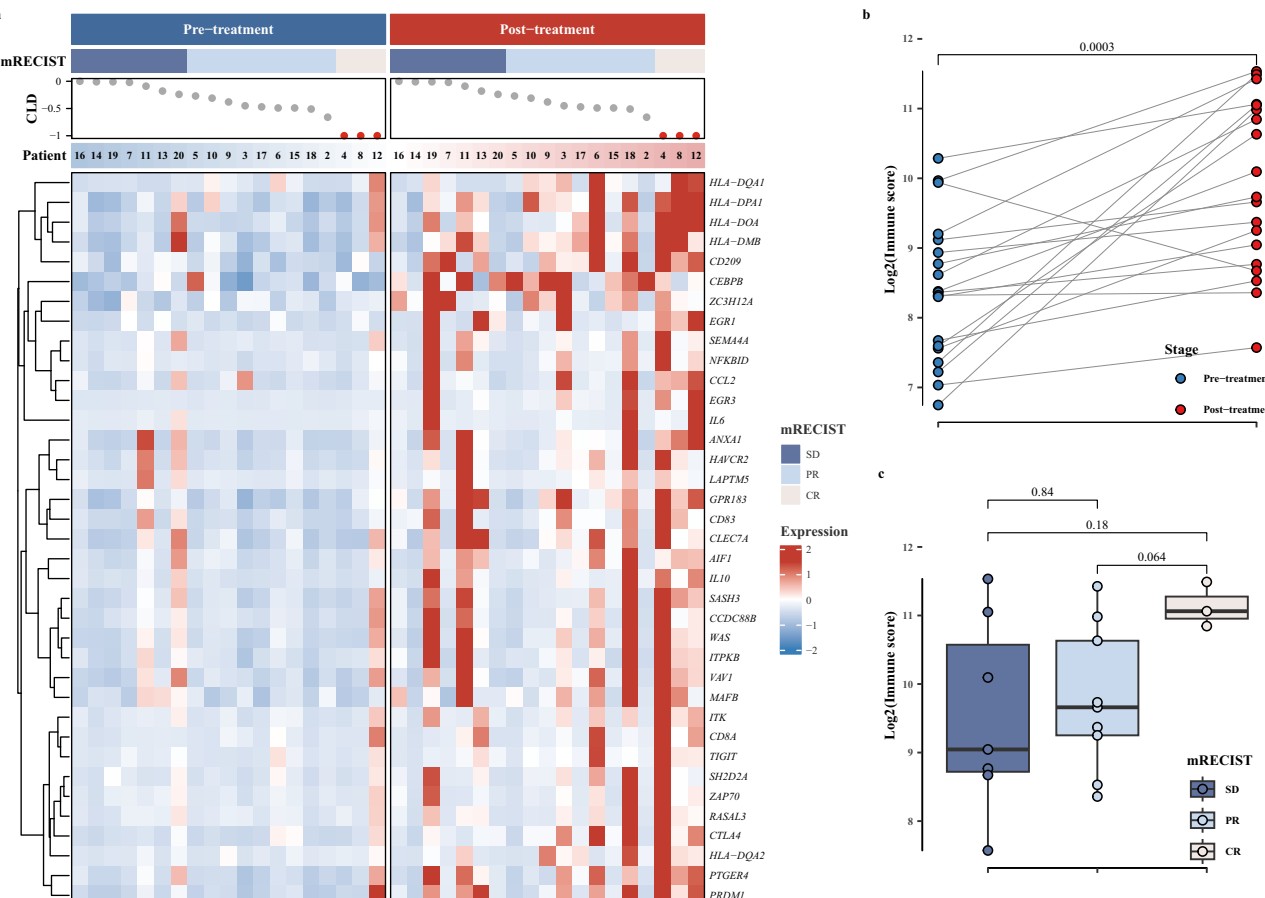

**Fig. 5 | Expression differences of T-cell activation-related genes between pre- and post-neoadjuvant therapy. a** Heatmap representation of T cell activates related genes expression in patients at pre- and post-neoadjuvant therapy; **b** comparison of immune scores of T cell activates related genes between pre- and post-neoadjuvant therapy ($n = 19$); **c** Comparison of immune scores of T cell activates related genes across mRECIST response categories. The mRECIST categories were defined as SD (Stable Disease) ($n = 7$), PR (Partial Response) ($n = 9$), and CR (complete response) ($n = 3$); CLD represents the assessment of the change in the longest diameter according to mRECIST criteria. Patient bar labeled by patient ID. The heatmap illustrates the scaled values derived from the gene expression data, ranging from blue to red, indicating increasing values. Blue represents patients in pre-neoadjuvant therapy, and red represents patients at post-neoadjuvant therapy. Boxplot accompanied with jittered points illustrates the distribution of log2 transformed immune scores for each patient. Within the box, horizontal line represents the median, and box represents 25th and 75th percentile. Whiskers are calculated with the formula median ± 1.5 × interquartile range. Paired $t$ tests are conducted between pre- and post-treatment data, whereas pairwise comparisons across different mRECIST categories are assessed using the Wilcoxon rank-sum method. Exact two-sided p-values, with significance levels ($p$ value < 0.05), are provided above the respective comparisons. Source data are provided as a Source Data file.

melanoma, compared to the adjuvant ICIs-only adjuvant therapy, adding neoadjuvant ICIs significantly improves the 3-year distant disease-free survival[37].

When combining radiotherapy with ICIs as neoadjuvant therapy, we are expecting not only a combining, but a synergistic therapeutic effect. We hypothesized that radiotherapy could enhance the antitumor immune response associated with the immune checkpoint blockade in the neoadjuvant setting. After the neoadjuvant therapy, we observed a significant increase in lymphocyte infiltration compared to the paired pre-neoadjuvant biopsy samples of the same patients (supplementary fig. 3), this easily accessible pathological finding may be an indication of the enhanced antitumor immune response.

Biomarker analysis based on RNA-seq data revealed a remarkably holistic immune activation from neoadjuvant therapy of tislelizumab plus SBRT, including the enhancement of adaptive and innate immunity. Remarkably, all CD8 T cell subtypes, most of CD4 T cell subtypes, and the states of T-cell activation exhibited a marked increase in scores, indicating a pivotal role of the neoadjuvant tislelizumab plus radiotherapy in augmenting T cell-mediated immune responses.

T cell immunity requires recognition of antigens in the context of MHC class I and class II proteins by CD8[+] and CD4[+] T cells, respectively.

Restoration of MHC-I expression, a common immune escape mechanism, is one of the processes by which radiotherapy might enhance response to immune checkpoint blockade[38]. We found that the neoadjuvant therapy widely enhanced the expression of HLA class I and II genes, esp. the class II genes; and the enhancement was particularly notable in patients with radiographic tumor response. Combined with the elevated expression levels of genes related to dendritic cells and Tfh cells post-treatment, these data indicated a potential correlation between the treatment response and treatment-induced enhancement of antigen presentation and processing and may partially explain the immune activation induced by treatment.

Tumor-reactive T cells emerged as key determinants of cancer immunity and response to immune checkpoint blockade (ICB) based therapies and were usually characterized by distinct phenotypes, such as tissue-resident memory T (Trm) cells which are early responders to pre-surgical cancer immunotherapy marked by ITGAE, ZNF683, ITGA1, and CXCR6[39,40], dysfunctional or exhausted T cells with remained cytotoxicity and proliferation capability, and high clonality marked by high expression levels of CXCL13, ENTPD1, and checkpoints[41–44]. The RNA-seq analyses in our data showed a significant upregulation of the gene signatures of tumor-reactive T cells and cytotoxic cells following

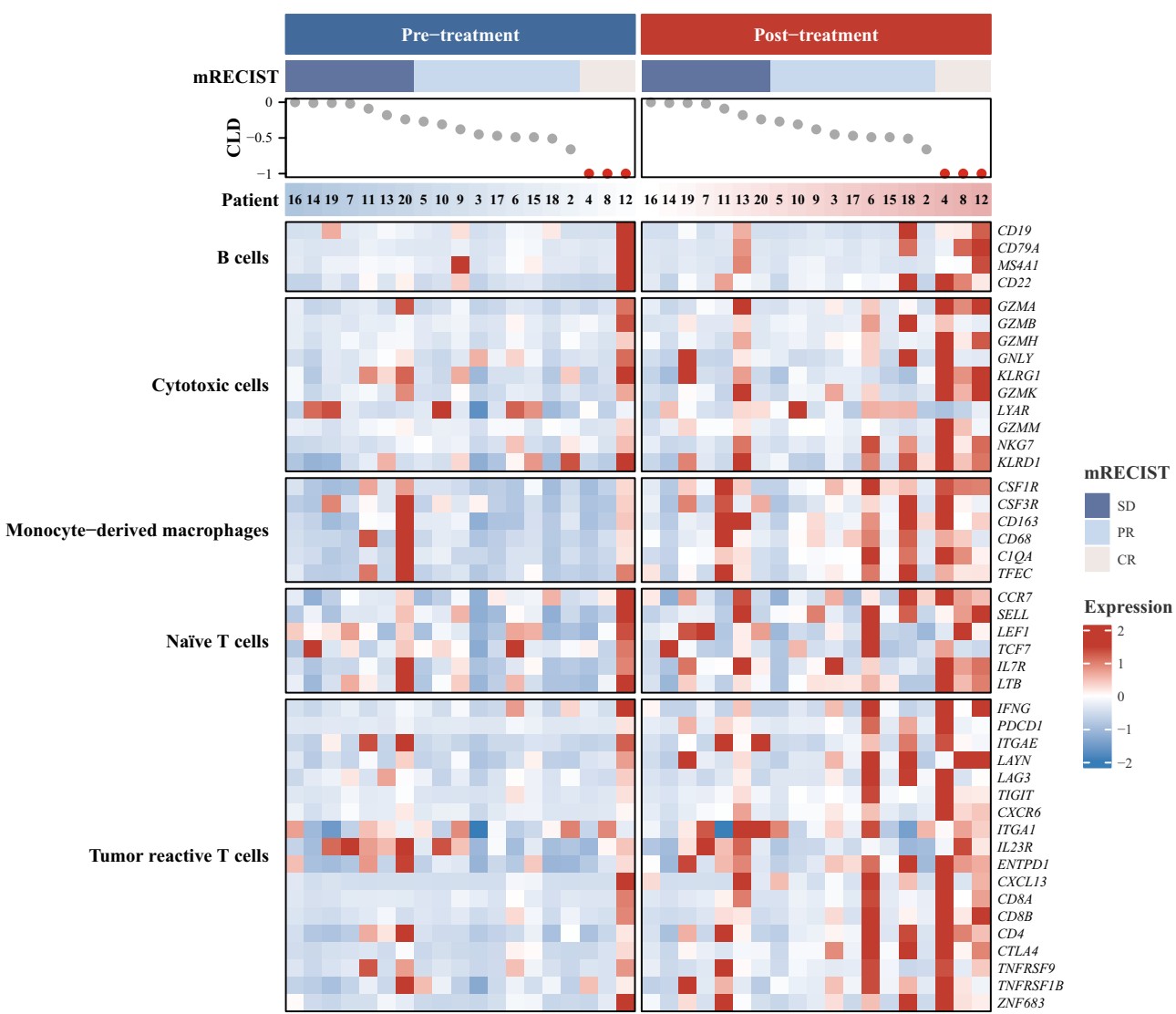

**Fig. 6 | Heatmap representation of published immune cells gene signatures expression at pre- and post-treatment (*n* = 19).** The mRECIST categories were defined as SD (Stable Disease), PR (Partial Response), and CR (Complete Response); CLD represents the assessment of the change in the longest diameter according to mRECIST criteria. Patient bar labeled by patient ID. The heatmap illustrates the scaled values derived from the gene expression data, ranging from blue to red, indicating increasing values. Blue represents patients at pre-treatment, and red represents patients at post-treatment. Source data are provided as a Source Data file.

the treatment. Noteworthy, they were more significant in patients with deeper radiographic tumor responses. These findings suggest a notable activation of those effective T cell components following the neoadjuvant therapy. Additionally, the mobilization and recruitment of new TCR clonotypes to the tumor site from the periphery or draining lymph nodes, in a phenomenon termed "clonal replacement" detected at 4–9 weeks after immunotherapy, has been identified as a key mechanism of response to ICB[41,43]. These newly emerging clones had a preferential exhausted T (Tex) cell state with high clonality, and both bearing signatures of activated cytotoxic T cells[39,45]. In consistent with this, we did observe a conspicuously high fraction of newly infiltrating TCRs which are mainly encompassed by large clonotypes post-treatment. These indicate that neoadjuvant therapy of tislelizumab plus SBRT may act systemically facilitate the clonal expansion and turnover of effective T cells through activation and expansion outside of the TME of antitumor T cells recruited to the tumor site. In addition, high proportion of hyperexpanded TCR clones could be found among TCR clonotypes which could be detected both pre- and post-treatment, suggesting the co-persistence of local activity of T cells.

Several limitations exist in this study. Firstly, the study was limited by the nature of its study design, with a single center and a relatively small sample size. With a relatively short duration of postoperative adjuvant treatment and study follow-up, whether this neoadjuvant setting can reduce the incidence of recurrence and finally improve the OS of early-stage HCC patients remains an unanswered question and needs extended follow-up. Also, as a single-arm study, there was not a control arm with mono-anti-PD-1 neoadjuvant therapy to compare with, so currently we cannot prove the synergistic immune-enhancing effect from radiotherapy.

In summary, our results showed that the neoadjuvant SBRT combined with anti-PD-1 was safe and well tolerated, it achieved clinically promising radiographic and pathological tumor responses, and no surgical cancellation or delay occurred due to any TRAEs or disease progression; the surgical resection for HCC after this neoadjuvant therapy was also safe, and an enhanced anti-tumoral immune response was observed after this combination neoadjuvant therapy. Collectively, these pilot findings warrant further clinical trials to explore the application of neoadjuvant ICI(s) + radiotherapy in HCC.

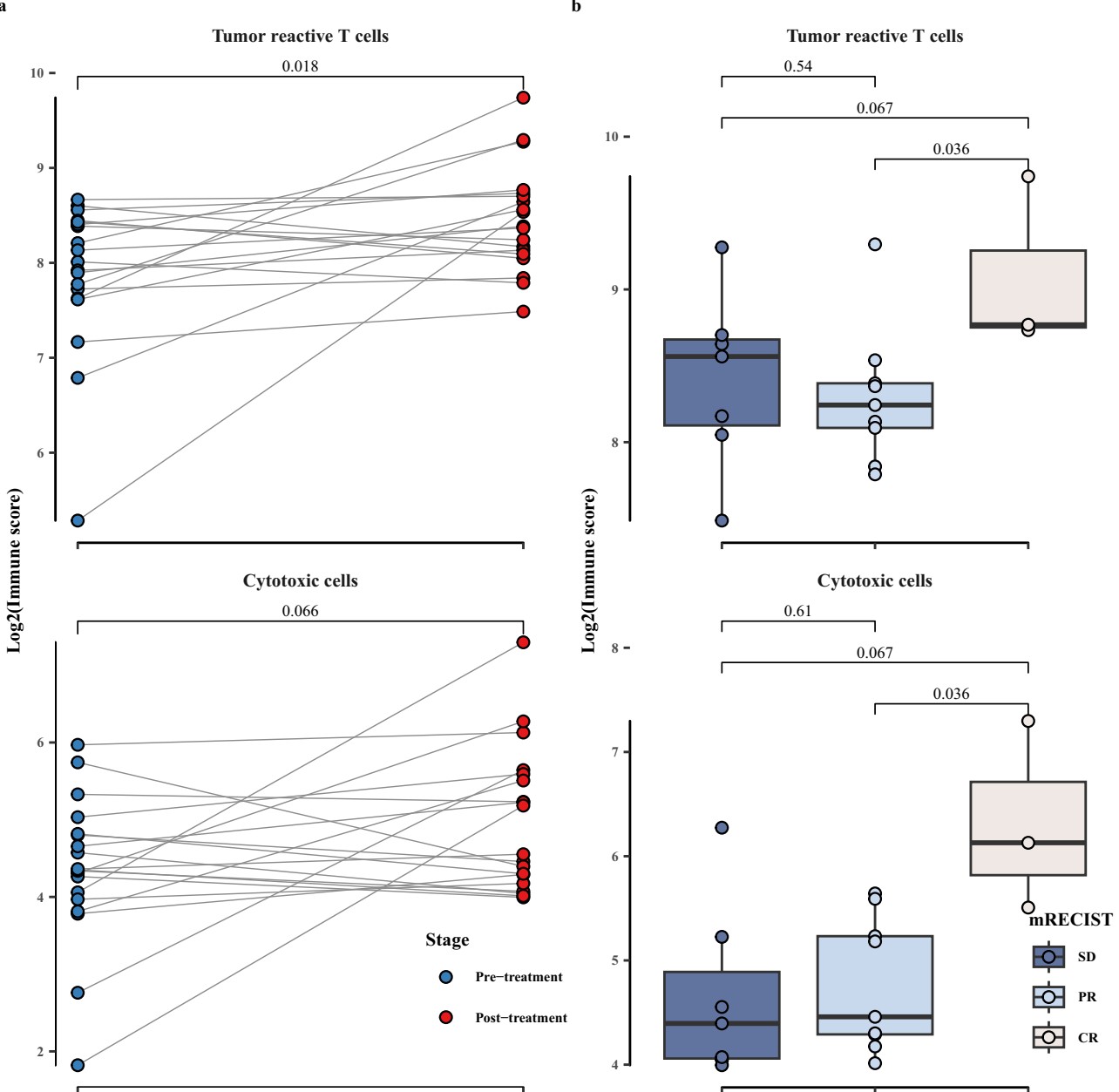

**Fig. 7 | Immune scores of published immune cells gene signatures.** Comparison of immune scores of published immune cells gene signatures between pre- and post-neoadjuvant therapy (**a**) ($n = 19$) and across mRECIST response categories (**b**) SD ($n = 7$): stable disease, PR ($n = 9$): partial response, and CR ($n = 3$): complete response; blue represents patients at pre-neoadjuvant therapy and red represents patients at post-neoadjuvant therapy. Points show the log2 transformed immune scores for each patient. Boxplot accompanied with jittered points illustrates the distribution of log2 transformed immune scores for each patient. Within the box,

horizontal line represents median, and box represents 25th and 75th percentile. Whiskers are calculated with the formula median ± 1.5 × interquartile range. Paired t-tests are conducted between pre- and post-treatment data, whereas pairwise comparisons across different mRECIST categories are assessed using the Wilcoxon rank-sum method. Exact two-sided $p$ values, with significance levels ($p$ value < 0.05), are provided above the respective comparisons. Source data are provided as a Source Data file.

## Methods

This trial was approved by the Ethics Committee of Shandong Cancer Hospital and Institute (SDZLEC2022-021-01). It was conducted in accordance with the International Council for Harmonisation of Technical Requirements for Pharmaceuticals for Human Use and the Declaration of Helsinki. All patients provided written informed consent before participating in this trial.

### Study design and participants

Notable-HCC (NCT05185531) is a single-center phase 1b study of neoadjuvant SBRT plus tislelizumab in patients with early-stage resectable

HCC. This trial was registered with ClinicalTrial.gov, NCT05185531, on 11 January 2022, and is ongoing but closed to accrual. The study protocol has been published previously[46] and a synopsis is also available in the Supplementary Information file. There is a lack of data on the treatment of neoadjuvant anti-PD-1 monoclonal antibody plus SBRT in early-stage resectable HCC patients in previous studies. Based on the feasibility of enrollment, 20 patients will be enrolled to evaluate the preliminary efficacy. No formal hypothesis testing will be performed in the efficacy evaluation. In the other two pilot, early-stage studies about the neoadjuvant therapy of ICI(s) in HCC, the sample size was 21 and 30, respectively.

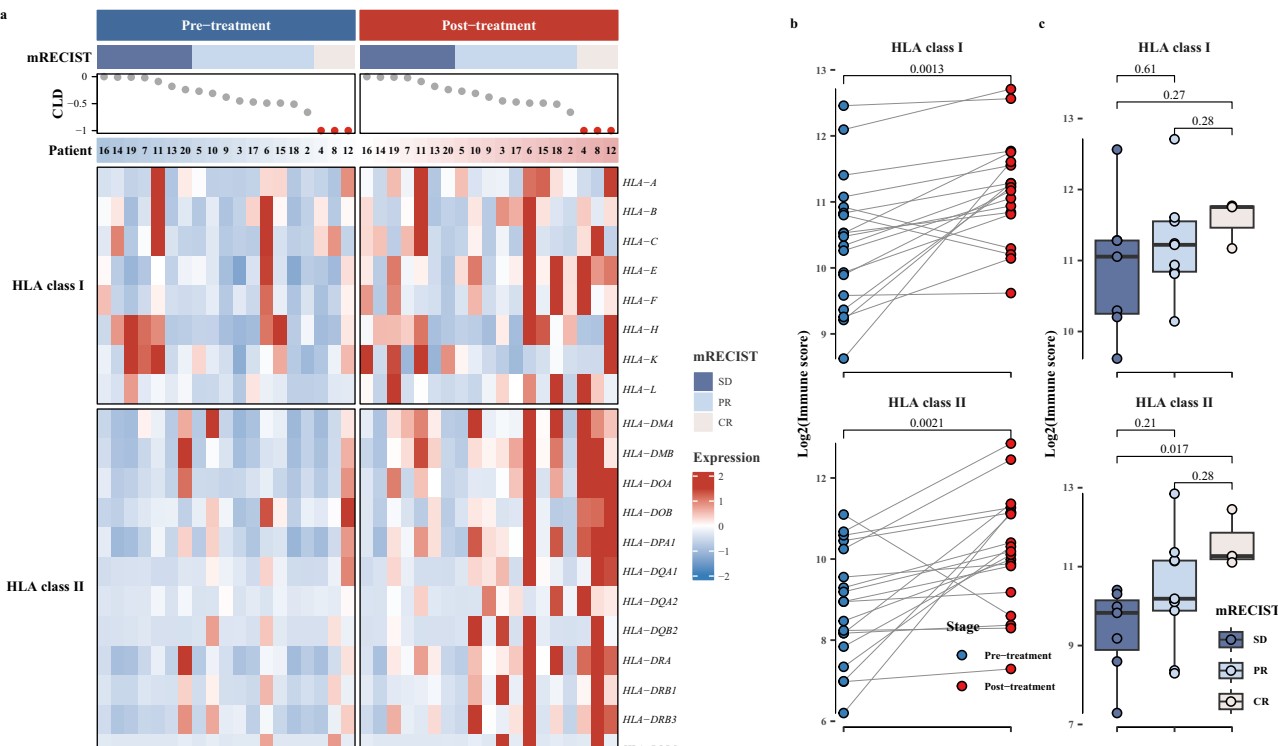

**Fig. 8 | HLA genes expression changes post-neoadjuvant therapy. a** Heatmap representation of HLA genes expression in patients at pre- and post-neoadjuvant therapy; **b** comparison of immune scores of HLA genes between pre- and post-neoadjuvant therapy (n = 19); **c** comparison of immune scores of HLA genes across mRECIST response categories. The mRECIST categories were defined as SD (stable disease) (n = 7), PR (partial response) (n = 9), and CR (complete response) (n = 3); CLD represents the assessment of the change in the longest diameter according to mRECIST criteria. Patient bar labeled by patient ID. The heatmap illustrates the scaled values derived from the gene expression data, ranging from blue to red, indicating increasing values. Blue represents patients at pre-treatment, and red represents patients at post-treatment. Points show the log2 transformed immune scores for each patient. Boxplot accompanied with jittered points illustrates the distribution of log2 transformed immune scores for each patient. Within the box, horizontal line represents median, and box represents 25th and 75th percentile. Whiskers are calculated with the formula median ± 1.5 × interquartile range. Paired t-tests are conducted between pre- and post-treatment data, whereas pairwise comparisons across different mRECIST categories are assessed using the Wilcoxon rank-sum method. Exact two-sided p-values, with significance levels (p value < 0.05), are provided above the respective comparisons. Source data are provided as a Source Data file.

The study started on 1 March 2022, the first patient enrolment date was March 22, 2022, and the last patient enrolment date was July 17, 2023. Eligible patients were aged ≥18, with histologically or radiographic confirmed, resectable HCC of BCLC stage 0 to A; the patients' Eastern Cooperative Oncology Group performance status (ECOG PS) were of 0 or 1, and had at least one measurable lesion by CT-scan or MRI defined by RECIST v1.1 and HCC-specific mRECIST, and overall Child-Pugh class were A. Patients were enrolled regardless of the underlying cause of HCC: (1) patients with active HBV infection (HBV DNA < 2000 IU/ mL during screening) were eligible if they initiated anti-HBV treatment at least 14 days prior to SBRT and were willingness to continue anti-HBV treatment during the study (per local standard of care; eg, entecavir); (2) for patients with HCV, either with resolved infection (as evidenced by detectable antibody and negative viral load) or chronic infection (as evidenced by detectable HCV RNA), were eligible. HCC patients who presented with chronic viral hepatitis, baseline bone marrow suppression, or liver dysfunction were eligible for enrollment if they demonstrated a positive response to symptomatic treatment and were assessed by the investigators as being able to tolerate the neoadjuvant treatment and subsequent hepatic resection. The corresponding indicators needed to be dynamically monitored throughout the course of treatment.

Patients were excluded if they had a known additional malignancy that was progressing or requiring active treatment. Patients could not receive any prior systemic anticancer treatment (including an anti-PD-(L)1 or anti-CTLA-4 antibody) for HCC, or underwent prior orthotopic liver transplantation. Patients with prior abdominal irradiation, and any major surgery within the 3 weeks prior to enrolment were excluded.

## Procedures

Eligible patients received three fractions of 8 Gy SBRT on day 1, 3, and 5. Both CT and MRI simulations, complemented with abdominal compression and 4DCT were performed to manage respiratory movement and accurately localize the target area. The target delineations and plan evaluations of all 20 patients receiving SBRT are available in the Source data file. The Elekta Unity MRI-linac system enables visualization of all anatomical changes during the course of radiotherapy, and hence adapt the treatment plan accordingly (supplementary movie 1). The radiotherapy process was meticulously overseen by a multidisciplinary quality assurance team.

A total of two cycles of tislelizumab at a dose of 200 mg by intravenous infusion in a 21-day cycle was planned for each patient: the first dose of tislelizumab was administered immediately after SBRT on day 1; the second dose was administered on day 22 (the first day of week 4, ±3 days). Curative liver resection of HCC was scheduled on day 50 (the first day of week 8, ±7 days). In the adjuvant setting, starting four weeks after the curative resection, patients received 200 mg of tislelizumab intravenously every 3 weeks for up to 1 year or until disease progression or intolerable toxicity (supplementary Fig. 1).

CT scans at chest, abdomen, and pelvis, and contrast-enhanced magnetic resonance imaging (MRI) scans at the liver during screening were performed to obtain baseline tumor imaging. Tumor responses

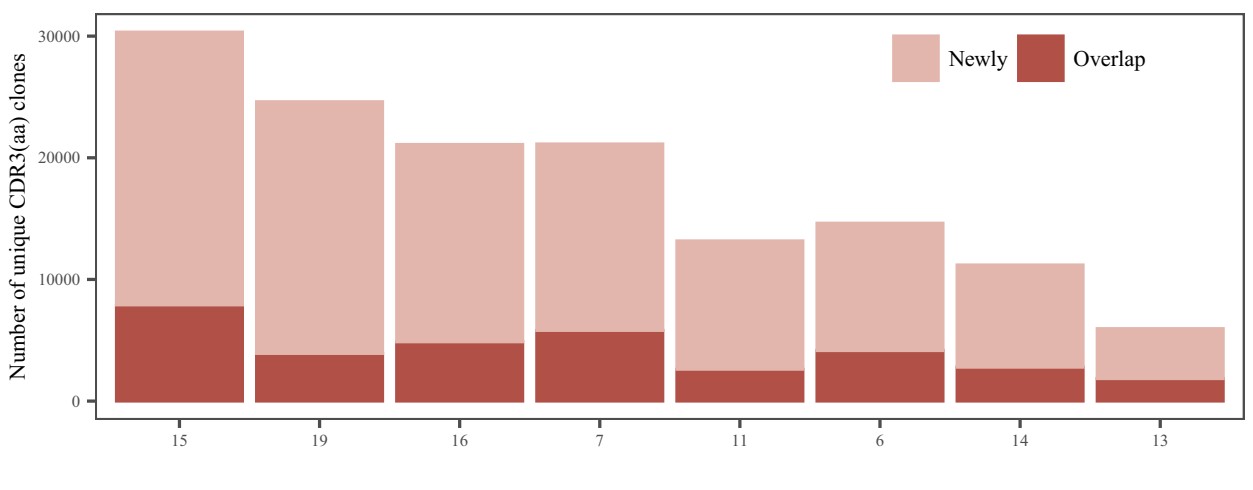

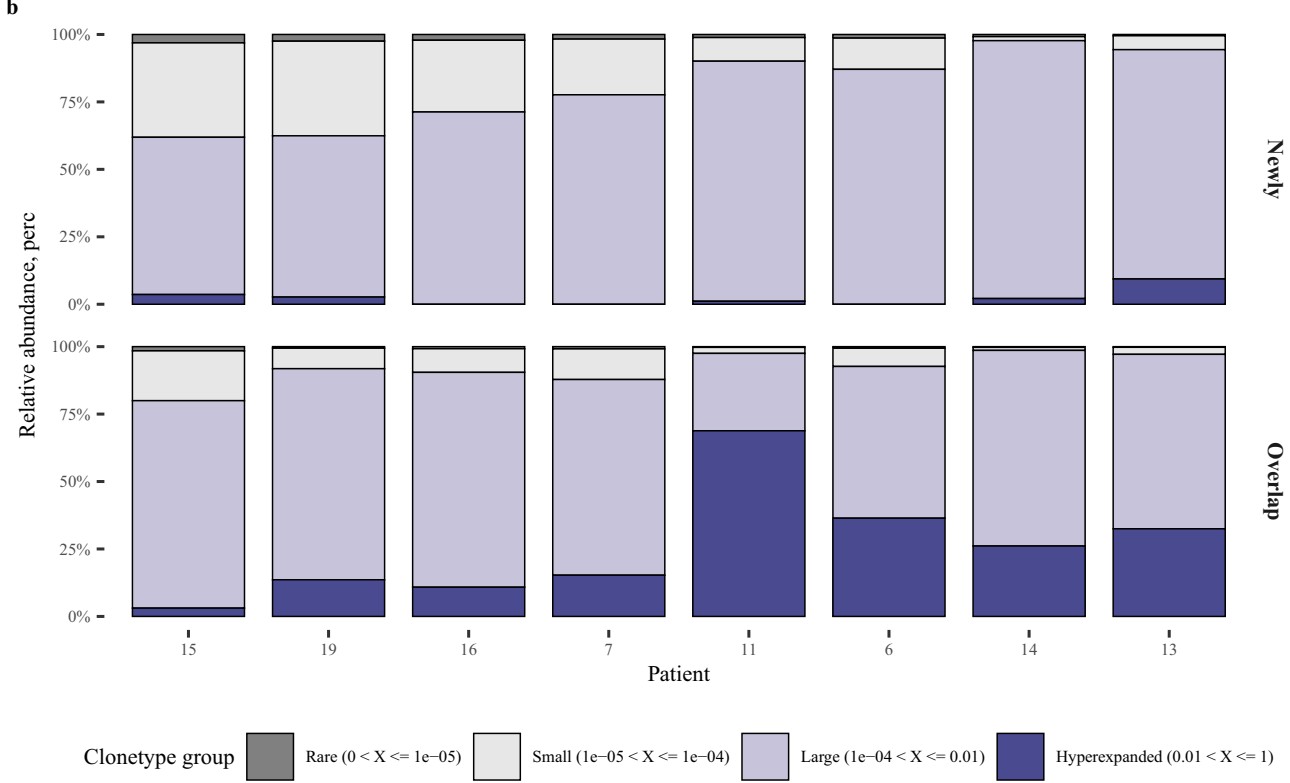

**Fig. 9 | TCR clonotypes changes post-neoadjuvant therapy (n = 8).** Number (**a**) and frequency (**b**) between newly generated and overlapping CDR3 (aa) clonotypes "Newly" represents newly generated clonotypes after treatment, and "overlap" represents overlapping clonotypes present in both pre- and post-neoadjuvant therapy. Hyperexpanded: clonotypes with frequency >1%; Large: clonotypes with frequency ranging from 0.01% to 1%; Small: clonotypes with frequency from 0.001% to 0.01%; Rare: clonotypes with frequency <0.001%. Source data are provided as a Source Data file.

following neoadjuvant therapy were evaluated using MRI prior to the surgery procedure; subsequently, patients underwent MRI scans 4-week post-surgery, followed by radiographic assessments every 3 months. For patients with contraindications to MRI, triple-phase CT of the liver was considered an acceptable alternative.

Samples from pre-treatment tumor biopsies and post-surgical resected tissues, as well as the peripheral blood mononuclear cells (PBMCs) collected at baseline and after the neoadjuvant therapies, were snap-frozen and stored at the biobank for biomarker analyses.

Safety and response to treatment after the neoadjuvant therapies were assessed before surgery. Tumor responses were measured per RECIST v1.1 and HCC-specific mRECIST criteria, including CR, PR, and ORR. Hepatic resection was performed as per the standard of care. A multidisciplinary team (MDT) assessed the patients' conditions and decided on the surgical protocol according to guidelines, experience, and MDT discussions. After resection, participants were assessed every 3 months (±7 days) and thereafter to collect information regarding disease status and survival. Long-term follow-up would continue for a total of 2 years for each patient.

Safety was monitored continuously throughout the trial. Adverse events (TRAEs) were assessed using the Common Terminology Criteria for Adverse Events (CTCAE) version 5.0. TRAEs included those events

considered by the investigator to be related to study treatment or with missing assessment of the causal relationship. The incidence of TRAEs was reported as a number (and percentage). A patient was counted only once by the highest severity grade. In cases where laboratory investigations showed abnormal values at baseline, any deterioration exceeding two grades will be further explored and reported.

To assess the impact of neoadjuvant therapy on surgical safety, we retrospectively analyzed all consecutive HCC patients of BCLC 0-A stage in our center, who underwent upfront curative resection during the simultaneous period as the trial was ongoing. Patients' demographics, surgery characteristics and incidence of surgical complications were reviewed in this retrospective cohort and compared with the patients enrolled in the trial.

### Outcomes and endpoints

Primary endpoints were the number of patients experiencing a surgery delay over 6 weeks (calculated from the planned date of surgery on day 50), ORR after the neoadjuvant therapy according to the RECIST v1.1 and mRECIST criteria, pathological response rates, and the safety and tolerability of the combination neoadjuvant therapy with SBRT+ tislelizumab, as well as the adjuvant therapy with tislelizumab. Secondary endpoints were 1-, 3- and 5-year DFS and OS rates assessed every 3 or 6 months after hepatic resection. Exploratory endpoints included patients' immune response, incidence of surgical complications, and mortality rate.

### Biomarker analysis

**Bulk RNA sequencing.** Pre- and post-treatment tumor tissues from 19 patients were subjected to RNA-seq analysis. Following RNA extraction, purification, and reverse transcription, libraries were prepared using TruSeq® RNA Exome Library Preparation Kit (Illumina) following the manufacturer's instructions. After library construction, library quality control was analyzed by Qubit 3.0 fluorometer dsDNA HS Assay and Agilent BioAnalyzer (Agilent). About 100 ng total RNA with DV200 > 30% was utilized as input total RNA. The RNA was fragmented into smaller fragments, cDNA was synthesized from the cleaved RNA fragments during first and second-strand synthesis, and adapters were subsequently ligated to the resulting double-stranded cDNA fragments. The coding regions of the transcriptome were then captured from this library using sequence-specific probes to create the final library. Paired-end 150 bp sequencing was conducted using the Illumina Novaseq6000 platform in Sequanta Technologies Co., Ltd.

### Bulk RNA-seq data processing

FastQC v0.11.9 software assessed the quality of raw data from high-throughput sequencing platforms. Subsequently, fastp v0.20.1 was employed to trim potential sequencing adapters from the raw reads and eliminate low-quality and ambiguously based reads[47]. For RNA-seq data, alignment to the hg19 reference genome was conducted using STAR v2.7.8a to generate RNA alignment BAM files[48].

Gene expression quantification was performed on the RNA-seq data using RSEM v1.3.3 to obtain raw read counts[49]. Subsequent normalization of expression abundance was carried out using edgeR v3.28.1 in R language by Trimmed Mean of M-values (TMM) normalization method[50].

### TCR sequencing

Genomic DNA extracted from pre- and post-treatment tumor tissues of eight patients underwent integrity assessment via agarose gel electrophoresis and quantification using Qubit. Targeted multiplex PCR amplification of T cell receptor β-chain (TRB) regions employed the FAIR-SEQ® Human TRB VJ multiplex Kit, followed by introducing sequencing adapter sequences and library Indexes and subsequent

purification using magnetic beads for library preparation. After quality assessment of the library, paired-end 150 bp reads were generated by Novaseq6000 platform for comprehensive TCR (T cell receptor) repertoire profiling in Sequanta Technologies Co., Ltd.

### TCR data processing

Raw sequencing data underwent initial quality assessment using FastQC v0.11.9 for quality control purposes. Subsequently, trimming of low-quality reads or sequences was performed employing cutadapt with parameters set to "-U 30 --trim-n -q 25,25 -e 0.1 -m 20". The processed data were then aligned against TCR immune repertoire references utilizing MiXCR v3.0.13 software[51]. The software identified and quantified clonotypes with MiXCR workflow: align, assemble, and export. MiXCR also corrected PCR and sequencing errors in the TRB repertoires[51]. Finally, vdjtools software was utilized to convert the output from MiXCR into a format compatible with downstream analyses[52].

### Statistical analysis

A total of 20 patients were planned to be enrolled in this trial. All participants who complete at least one dose of tislelizumab and one fractions of SBRT will be included in the safety analysis (SAS). All participants in SAS who complete curative HCC resection will be included in the efficacy analysis (EAS). The baseline demographic and clinicopathological variables will be presented by descriptive analyses. RECIST 1.1/mRECIST response rates (CR, PR, and ORR) and pathological response rates (MPR [defined as residual tumor cells of 30% or fewer in the resected specimen], pCR [complete pathological response], etc) will be presented descriptively. Follow-up time and DFS/OS will be calculated from the day of HCC resection. Statistical analyses of clinical parameters were done with SPSS.

### Statistical analyses of RNA-seq and TCRseq

All downstream analyses and statistical comparisons were conducted using R software version 4.3.0. Single-sample Gene Set Enrichment Analysis (ssGSEA) of immune infiltration of bulk RNA-seq data was executed by GSVA package[53,54]. Gene sets related to T cell activation were derived from the Gene Ontology (GO) Term GO:0042110. Cell type identification within the RNA-seq data utilized previously established gene signatures for naïve and cytotoxic T cells, B cells, and monocyte-derived macrophages to quantify lymphocyte populations in tumor specimens before and after treatment. The genes of tumor-reactive T cell signature include CD8A, CD8B, CD4, CXCL13, PDCD1, CTLA4, LAG3, TIGIT, ENTPD1, TNFRSF9, TNFRSF1B, IFNG, LAYN, IL23R, ITGAE, ITGA1, CXCR6, and ZNF683. The immune scores were computed by summing the normalized counts of all signature genes comprising each cell type and then transformed by log2 scale. Statistical significance was assessed by paired t-test for data at paired pre- and post-treatment and the Wilcoxon rank-sum test for unpaired data across different mRECIST categories. TCR clonotype frequencies were statistically evaluated using the immunarch package[55]. Visualization of results was carried out using the ggplot2 package for generating graphical representations, and heatmaps were generated using the Complexheatmap package[56].

### Reporting summary

Further information on research design is available in the Nature Portfolio Reporting Summary linked to this article.

## Data availability

The raw RNA and TCR sequencing data generated in this paper have been deposited in the Genome Sequence Archive in the National Genomics Data Center, China National Center for Bioinformation, under accession code HRA006511. The sequencing data are available under controlled access due to data privacy laws related to patient

consent for data sharing, and the data should be used for research purposes only. Access can be obtained by completing the application form via GSA-Human System. For detailed guidance on making the data access request, see GSA-Human_Request_Guide_for_Users [https://ngdc.cncb.ac.cn/gsa-human/document/GSA-Human_Request_Guide_for_Users_us.pdf]. The approximate response time for accession requests is about 4 weeks, and access will be granted for one year. Clinical data are not publicly available due to involving patient privacy, but can be accessed from the corresponding author Lei Zhao (Email: drzhaolei@hotmail.com), upon request for 3 years; individual de-identified patient data will be shared for clinical study analyses. The study protocol is available in the Supplementary Information file. The remaining data are available in the manuscript, Supplementary Information, or Source Data file. Source data are provided in this paper.

## Code availability

All code for data analysis and visualization employed in this work is public available at: https://github.com/lzc19880909/hcc_neoadjuvant.

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

## Acknowledgements

The authors thank the participating patients and their families for their trust and contributions to the study. We also appreciate the overtime work of Sequanta Technologies, which guaranteed the completion of the biomarker test before the submission deadline. L.Z. is funded by the National Natural Science Fund of China (no. 81872400, 81472713, 81272375), The Key Research and Development Program of Shandong (Major Science & Technology Innovation Project) (2021SFGC0501), and Joint Innovation and Development Item of Shandong Natural Scientific Fund (ZR2021LZL008). BeiGene, Ltd. partly funded this study and provided tislelizumab but had no role in the study design or writing of the manuscript.

## Author contributions

L.Z. conceived and designed the project, applied for the funding, did the operations, drafted and revised the manuscript; Z. Li and J.L. drafted the manuscript, reviewed the medical record and did statistical analysis; Z.Li., B.Z., X.S., K.C., L.L., C.Z., P.S., and J.Z. did the operations; J.L. and J.Y. did the SBRT; Z. Li, J.L., Z. Liu, Z.C., Z.S., M.L., Y.Y. followed-up the patients; Z.M. evaluated the tumor responses.

## Competing interests

L.Z. is on the speakers' bureau for Bayer, MSD, AstraZeneca, Roche, BeiGene, Innovent, Junshi Biosciences, and Hengrui Medicine. The remaining authors declare no competing interests.
