## [Peer Review File · Nature Communications]

Neoadjuvant tislelizumab plus stereotactic body radiotherapy and adjuvant tislelizumab in early-stage resectable hepatocellular carcinoma: the Notable-HCC phase 1b trialREVIEWER COMMENTS

Reviewer #1 (Remarks to the Author): with expertise in clinical trial study design, biostatistics

This is a single-site single-arm Phase Ib trial of neoadjuvant tislelizumab with stereotactic body radiotherapy in patients with early-stage resectable hepatocellular carcinoma (HCC). In view of the single-arm design of this trial, the authors included a cohort of 203 HCC patients who underwent upfront curative resection in the same institution during the same period when the trial was ongoing.

Major comments

1. Several primary endpoints were listed for this trial, including (1) the number of patients experiencing a surgery delay over 6 weeks, (2 & 3) ORR after the neoadjuvant therapy based on RECIST and mRECIST criteria, (4) pathological response rates, and (5) the safety and tolerability of the combination neoadjuvant therapy with SBRT+tislelizumab and adjuvant therapy with tislelizumab. With such a list of primary endpoints, how was the sample size for this Phase Ib trial determined?
2. With reference to the primary endpoint (1) above, perhaps, it is not the “number” of patients but the “proportion/percentage” of patients who experienced a surgery delay over 6 weeks that is more informative. The “number” of patients without knowledge of the denominator is not informative or generalizable.
3. The authors included a cohort of 203 HCC patients treated with upfront curative resection in the same institution during the same period, however, this cohort did not serve to provide any important insights with regards to the primary endpoints (perhaps primary endpoint 5?) of this trial at all.
4. Figure 3, it will be good to include in the Figure caption what does the asterisk above the bar represent?
5. Figure 4, could the authors please state what do “2-SII” and “2-SIV” represent?
6. Line 399, it appears that there are only 11 bars that extended below the -30% mark for mRECIST in Figure 4.
7. Figure 5, it is difficult to read the RECIST (filled square) and mRECIST (hollow diamond) response markers on the figure. Is there a way to make the diamond clearer?
8. Line 418 and 420, could the authors please clarify what time-point is baseline in these

sentences?

9. Line 425, could the authors include in the statistical section how the median follow-up time of 4 months was estimated?

Minor comments

1. Please note that there are several typos where trial was typed as trail.
2. Please note that ICH changed its name to The International **Council** for Harmonisation of Technical Requirements for Pharmaceuticals for Human Use.

Reviewer #2 (Remarks to the Author): with expertise in HCC, therapy

Congratulations to the authors for designing and conducting this exciting and potentially impactful clinical trial. This is a single-arm pilot prospective clinical trial of patients with resectable HCC with preserved liver function who received 2 cycles of neoadjuvant anti-PD1 with 3 fractions of SBRT, followed by surgical resection, then 1 year of adjuvant anti-PD1. They found that this treatment was safe, well tolerated, did not preclude surgical resection, and led to impressive radiographic and pathologic response rates as well as anti-tumor immunity. These findings are exciting and justify the design of a multicenter phase II/III trial. I have some general comments aimed at improving the manuscript.

- Consider adjusting the title to reflect perioperative tislelizumab, not just neoadjuvant
- There are many “primary” endpoints. Please comment. How was the primary endpoint of 6 weeks determined? It was “easily” met. Therefore, how was the sample size calculated.
- The manuscript is quite long and each section is wordy (although well written). Please ensure that it meets the journal’s formatting requirements.
- In particular, the abstract could be more succinct at times.
- Comparing the postoperative data to retrospective data was not mentioned in the methods. This could probably be presented in a supplemental table and not use an entire paragraph
- It may be helpful to reference similar work being conducted on the combination of SBRT/ICI in the neoadjuvant treatment of other tumors (e.g. RCC, NSCLC, melanoma)

Reviewer #3 (Remarks to the Author): with expertise in HCC, therapy

Zhongchao Li and his colleagues conducted a study on neoadjuvant therapy for early-stage HCC. In this study, they were the first to explore a novel adjuvant treatment regimen, which is SBRT combined with PD-1 monoclonal antibody. The preliminary results of this study are satisfactory, the ORR was 63.2% and the DCR was 100% per mRECIST. In particular, no patient progressed during neoadjuvant therapy and surgical planning was compromised due to AE. In addition, the authors performed RNA sequencing of pre- and post-treatment tumor tissue in 19 patients, as well as TCR sequencing in 8 patients. With tissue sequencing, the authors found that neoadjuvant therapy effectively activated the immune system with upregulation of T cell activation-related genes, enhanced HLA expression and a higher phenotype of newly generated TCR clonotypes. Overall, the study was well designed and executed, summarizing the clinical effects of neoadjuvant therapy and providing a more in-depth analysis of clinical phenomena at the genetic level. However, I still have some questions and believe that minor revisions are required.

1. Early-stage HCC has a relatively favorable prognosis, and more explanation is needed as to the rationale for choosing early-stage HCC (3 of which were BCLC stage 0) as the study population, why not intermediate or late stage?

2. One patient received radiofrequency ablation, other 19 patients achieved curative R0 resection. When calculating the ORR and DCR, why not use the 20 people enrolled in the group? In Figure 4, the results for 20 patients are shown?

3. The study included 4 cases of recurrent tumors; does this have an impact on the results? Two of these were recurrent within two years of initial resection. It is generally accepted that early recurrence (within 2 years) is associated with the primary tumor and in some way implies higher aggressiveness.

4. Changes in immune-activated T cells before and after treatment were clearly observed in CR patient number 4 and 8 (Fig. 8), but not much in number 12; does this suggest that the initial immune status prior to treatment also has an impact on the treatment outcome?

5. In lines 528-554 of the discussion section, the authors mention disease progression during

neoadjuvant therapy as a concern and use 2 studies as examples. In lines 552-554, the authors claim that “the combination of a local-regional therapy to ICIs can significantly improve the local control rate and avoid the cancellation of curative resection due to tumor progression after neoadjuvant therapy”. Although no tumor progression was observed in this study, this conclusion seems inappropriate, and the authors then analyze the limitations of the study as well.

6. The author repeats many of the same items in the discussion section as in the results section, which could be considered appropriately refined.

7. Is postoperative adjuvant PD-1 monoclonal antibody necessary for patients with pCR?

8. Further description of MHC, HLA, and TCR clonotypes and relationship to immunization and treatment is recommended rather than simply describing the results.

9. The activation of immune cells in tumor tissues before and after treatment, and the regulation of related genes is one of the features of this study, which deserves a more in-depth analysis in the discussion section.

10. There is a formatting problem with the Trial design in Figure 1 that prevents the entire process from being seen.

Reviewer #4 (Remarks to the Author): with expertise in HCC, therapy

This is a scientifically sound and well-written paper.

Please elaborate more on the design to assess and account for adverse events (rolling 6, 3+3?)

Please reference the Himalaya trial and the Imbrave150 trials in the introduction and elaborate more on the rationale for using tislelizumab in your trial

Please reference other ongoing and published studies of neoadjuvant RT+/-IO for HCC (work by Ted Hong for example)

REVIEWER COMMENTS

Reviewer #1 (Remarks to the Author): with expertise in clinical trial study design, biostatistics

This is a single-site single-arm Phase Ib trial of neoadjuvant tislelizumab with stereotactic body radiotherapy in patients with early-stage resectable hepatocellular carcinoma (HCC). In view of the single-arm design of this trial, the authors included a cohort of 203 HCC patients who underwent upfront curative resection in the same institution during the same period when the trial was ongoing.

Major comments

1. Several primary endpoints were listed for this trial, including (1) the number of patients experiencing a surgery delay over 6 weeks, (2 & 3) ORR after the neoadjuvant therapy based on RECIST and mRECIST criteria, (4) pathological response rates, and (5) the safety and tolerability of the combination neoadjuvant therapy with SBRT+tislelizumab and adjuvant therapy with tislelizumab. With such a list of primary endpoints, how was the sample size for this Phase Ib trial determined?

Response:

We appreciate the reviewer for pointing out this, which is a pivotal question in the designing of clinical trials. This study is a non-hypothesis-driven observational study.

Our primary focus for this study is the rate (and number) of patients who experiencing a surgery delay over 6 weeks, which we have designated as the primary endpoint. The other endpoints are observational clinical valuables, due to its nature of the single-armed design, there were no statistical comparisons of these observational valuables.

To avoid confusion, we have classified them as secondary endpoints. We have made corresponding modification in the main text. This modification does not affect the interpretation of the results.

In terms of the sample size calculation, there is a lack of data on the treatment of neoadjuvant anti-PD1 monoclonal antibody plus SBRT in early-stage resectable HCC patients in previous studies. Based on the feasibility of enrollment, about 20 patients will be enrolled to evaluate the preliminary efficacy. No formal hypothesis testing will be performed in the efficacy evaluation.

In other two pilot, early-stage studies about the neoadjuvant therapy of ICI(s) in HCC, the sample size was 21¹ and 30², respectively. Following the pilot study to confirm the safety of neoadjuvant therapy with ICI(s) +/- SBRT in early-stage HCC, as we pointed out in the “Discussion”: “Further studies with larger sample sizes, incorporating a control group such as immune monotherapy, are warranted to validate the clinical benefits of this combination therapy.”

2. With reference to the primary endpoint (1) above, perhaps, it is not the “number” of patients but the “proportion/percentage” of patients who experienced a surgery delay over 6 weeks that is more informative. The “number” of patients without knowledge of the denominator is not informative or generalizable.

Response:

According to the reviewer’s opinion, we revised the “Outcomes and endpoints” section in the “Methods”, and set both the “number” and the “percentage” of patients who experiencing surgery delay over 6 weeks as the primary endpoints; we also revised the “Surgeries after neoadjuvant SBRT+ anti-PD-1” section in “Results” and

presented data of both numbers and percentages. Now the result is more informative and straightforward.

3. The authors included a cohort of 203 HCC patients treated with upfront curative resection in the same institution during the same period, however, this cohort did not serve to provide any important insights with regards to the primary endpoints (perhaps primary endpoint 5?) of this trial at all.

Response:

As what the reviewer has understood, our original intention of the inclusion of this cohort was to prove the safety of curative HCC resection after the neoadjuvant therapy of anti-PD-1 plus SBRT.

As what we pointed out in the “Discussion”, one of the general concerns of neoadjuvant therapy is “severe AE of neoadjuvant therapy can ... increase the risk of post-operative morbidity”. By comparing the incidences of postoperative complications and the (peri)-operative variables between this cohort of 203 patients and 19 patients who experienced neoadjuvant therapy, we intended to show that the

tested neoadjuvant therapy of tislelizumab plus SBRT did not increase the surgical difficulty and the risk of complications in the following HCC resection.

We thank the reviewer for pointing out the limitation of this cohort. Taking together the suggestions from other reviewers and the editors, we revised the manuscript and put the data and the discussion related to this cohort to a supplementary document.

4. Figure 3, it will be good to include in the Figure caption what does the asterisk above the bar represent?

Response:

Revised according to the requirement. The asterisk meant that the surgery delay in this patient was due to COVID-19 infection or quarantine policy.

5. Figure 4, could the authors please state what do “2-SII” and “2-SIV” represent?

Response:

By the reviewer’s question, we realized the label of “2-SII” and “2-SIV” were indeed difficult to understand, esp. for readers who are not liver surgeons.

“S” is the abbreviation for “segment”. According to the Couinaud liver segmentation, human liver is anatomically divided into 8 segments, which was first described by the French Surgeon Claude Couinaud in 1957. So “2-SII” refers to the tumor in segment II of patient No. 2, and “2-SIV” refers to the tumor in segment IV of patient No.2.

To make figure 4 easily understood and avoid confusion, we have modified the labeling of patient No. 2 on the X-axis; we used the full spelling of “segment” to replace the abbreviation of “S”, “II” and “IV” were changed to the subscript form.

6. Line 399, it appears that there are only 11 bars that extended below the -30% mark for mRECIST in Figure 4.

Response:

We appreciate your meticulous reviewing work. We made a mistake in plotting the mRECIST data of patient No. 5.

Per mRECIST criteria, the pre-treatment diameter of his tumor was 34.51mm, and 23.21mm post-treatment, so the tumor response was 33%, which reached PR. But it was mistakenly plotted as 27%. We have corrected the figure 4.

Representative MRI/CT images at the baseline and after the neoadjuvant therapy of all patients were presented in the Supplementary document 2.

Data of tumor response is very important for this study, and this mistake should have been avoided from the very beginning.

7. Figure 5, it is difficult to read the RECIST (filled square) and mRECIST (hollow diamond) response markers on the figure. Is there a way to make the diamond clearer?

Response:

We re-designed the layout of the figure, and avoided the overlapping of different symbols at the same time-points. We hope now all information is clearly presented, and the figure is more easy to read.

8. Line 418 and 420, could the authors please clarify what time-point is baseline in these sentences?

Response:

Here the “baseline” referred to the time-point when patients were just recruited and before the initiation of the neoadjuvant therapy. To avoid possible confusion, we revised the text and clearly stated it.

9. Line 425, could the authors include in the statistical section how the median follow-up time of 4 months was estimated?

Response:

We included the following explanation into the statistical section: “Follow-up time and DFS/OS will be calculated from the day of HCC resection”.

At data cutoff (Dec 1st, 2023), the follow-up times (in months) of 19 patients were 18.8, 15.7, 13.1, 9.9, 10.1, 9.1, 6.4, 6.7, 4.5, 4.0, 3.6, 3.8, 3.2, 2.8, 3.1, 2.9, 2.7, 2.2 and 2.6; the calculated 25% percentile, median and 75% percentile of the follow-up time were 2.9 months, 4.0 months and 9.9 months, respectively.

Minor comments

1. Please note that there are several typos where trial was typed as trail.

Response:

Three “trail(s)” had been corrected to “trial(s)”. These typos really should have been avoided in the first submission.

2. Please note that ICH changed its name to The International **Council** for Harmonisation of Technical Requirements for Pharmaceuticals for Human Use.

Response:

We appreciate the updated information provided by the reviewer, we revised the text according to the notification.

Reviewer #2 (Remarks to the Author): with expertise in HCC, therapy

Congratulations to the authors for designing and conducting this exciting and potentially impactful clinical trial. This is a single-arm pilot prospective clinical trial of patients with resectable HCC with preserved liver function who received 2 cycles of neoadjuvant anti-PD1 with 3 fractions of SBRT, followed by surgical resection, then 1 year of adjuvant anti-PD1. They found that this treatment was safe, well tolerated, did not preclude surgical resection, and led to impressive radiographic and

pathologic response rates as well as anti-tumor immunity. These findings are exciting and justify the design of a multicenter phase II/III trial. I have some general comments aimed at improving the manuscript.

Response:

We greatly appreciate the positive feedback from the reviewer on our work, esp. the encouragement for further phase II/III clinical trial.

- Consider adjusting the title to reflect perioperative tislelizumab, not just neoadjuvant

Response:

We re-wrote the title as:

Notable-HCC: A trial of neoadjuvant tislelizumab plus stereotactic body radiotherapy and adjuvant tislelizumab in early-stage resectable hepatocellular carcinoma

- There are many “primary” endpoints. Please comment. How was the primary endpoint of 6 weeks determined? It was “easily” met. Therefore, how was the sample size calculated.

Response:

For the primary endpoints and the sample size of the trial (another reviewer also asked this question):

We appreciate the reviewer for pointing out this, which is a pivotal question in the designing of clinical trials. This study is a non-hypothesis-driven observational study.

Our primary focus for this study is the rate (and number) of patients who experiencing a surgery delay over 6 weeks, which we have designated as the primary endpoint. The other endpoints are observational clinical valuables, due to its nature of the single-armed design, there were no statistical comparisons of these observational valuables.

To avoid confusion, we have classified them as secondary endpoints. We have made corresponding modification in the main text. This modification does not affect the interpretation of the results.

In terms of the sample size calculation, there is a lack of data on the treatment of neoadjuvant anti-PD1 monoclonal antibody plus SBRT in early-stage resectable HCC patients in previous studies. Based on the feasibility of enrollment, about 20 patients will be enrolled to evaluate the preliminary efficacy. No formal hypothesis testing will be performed in the efficacy evaluation.

For the setting of 6 weeks as the threshold of surgery delay:

When we designed this trial, we realized that the threshold to define the “surgery delay” can be relatively subjective, so we referred to the other trials of neoadjuvant therapy with ICI(s) in HCC, and there are only very few relevant publications in the literature.

In one such study, Kaseb and colleagues² reported that one patient had delayed surgery and a protocol deviation as a result of receiving six cycles of neoadjuvant nivolumab, but gave no criteria of surgery delay.

In another study, Marron and colleagues¹ included the delay of surgery as one of the secondary endpoints; the surgical resection was scheduled right after the second dose of cemiplimab (which had a ± 3 -day window), and the surgery delay was defined as more than 28 days following it. One patient had grade 3 pneumonitis during neoadjuvant therapy and required steroids treatment, which resulted in a delay of surgery by 2 weeks.

From our experience of the conversion surgery of HCC after the combination therapy including ICI(s)^{3, 4}, as well as from the literature, considering the risk of irAEs and their possible impact to surgery, we feel that liver resection right after ICI administration is challenging. The risk of first-onset irAEs is threefold higher during the first 4 weeks of treatment than between 4 weeks and the end of treatment⁵. In a large series of 122 cases, the median time from initiation of ICI to presentation of ICI-related myocarditis was 30 days, implying that most patients presented after receiving 1 or 2 doses ICI⁶. In our clinical practice, for HCC patients down-staged to resectable by the treatment including ICI, the surgery is commonly scheduled 4 weeks after the cessation of ICI. In our study, the neoadjuvant PD-1 was combined with SBRT, we also need to consider the influence of radiotherapy on the following surgery. For example, in the trial of neoadjuvant chemo-radiotherapy of locally advanced cancer of the oesophagus or oesophagogastric junction, the surgery was scheduled 4 to 6 weeks after the neoadjuvant therapy⁷.

We then referred to another ongoing trial of neoadjuvant ICIs in HCC⁸. In this study, rate of patients experiencing a surgery delay was included as one of the primary

endpoints, and it was defined as surgery delay to Day 89 or later; ipilimumab was administered once on Day 1, nivolumab was administered on Day 1 and Day 22 (± 3 days) for a total of two 21-day cycles (6 weeks of treatment, end on Day 42), so Day 89 was 47 days (2 days short for 7 weeks) after the ending of ICIs treatment cycle.

Referring to this trial, we set a delay of 6 weeks as the threshold of “surgery delay”.

In our manuscript, besides the overall number and percentage of surgery delay over 6 weeks, we also reported and discussed the individual “days between treatment start and surgery” of all 20 participants, we believe the detailed information will be helpful to objectively evaluating the endpoint of surgery delay in this study.

- The manuscript is quite long and each section is wordy (although well written).

Please ensure that it meets the journal’s formatting requirements.

Response:

we revised the manuscript and moved the data and the discussion related to the cohort of “203 HCC patients treated with upfront curative resection” to a supplementary document. According to another reviewer’s suggestion, we also refined the

“Discussion” section by deleting simple data results, which had been presented in the “Results” section. We will further refine the whole manuscript if the editor requires to do so.

- In particular, the abstract could be more succinct at times.

Response:

Nature Communications requires that the text of abstract should be no more than 200 words. We re-wrote the abstract and reduced the word counts to 199.

- Comparing the postoperative data to retrospective data was not mentioned in the methods. This could probably be presented in a supplemental table and not use an entire paragraph

Response:

We thank the reviewer for pointing out the limitation of the comparison between prospective and retrospective data. Taking together the suggestions from other reviewers and the editors, we revised the manuscript and put the data and the discussion related to this comparison to a supplementary document.

- It may be helpful to reference similar work being conducted on the combination of SBRT/ICI in the neoadjuvant treatment of other tumors (e.g. RCC, NSCLC, melanoma)

Response: We have quote Altorki and colleagues' study of neoadjuvant SBRT+ PD-L1 in non-small cell lung cancer, which is an important paper in the exploration in this direction. Similar pilot early-stage trials are still very limited, but have been reported in head and neck squamous cell carcinomas⁹, triple-negative breast cancer (TNBC)¹⁰. Just when we are revising this manuscript, another group in our hospital published a study of neoadjuvant radiation plus anti-PD-1 in esophageal squamous cell cancer¹¹. We thank the reviewer's suggestion, and we have cited these latest papers in the revised manuscript.

Reviewer #3 (Remarks to the Author): with expertise in HCC, therapy

Zhongchao Li and his colleagues conducted a study on neoadjuvant therapy for early-stage HCC. In this study, they were the first to explore a novel adjuvant treatment

regimen, which is SBRT combined with PD-1 monoclonal antibody. The preliminary results of this study are satisfactory, the ORR was 63.2% and the DCR was 100% per mRECIST. In particular, no patient progressed during neoadjuvant therapy and surgical planning was compromised due to AE. In addition, the authors performed RNA sequencing of pre- and post-treatment tumor tissue in 19 patients, as well as TCR sequencing in 8 patients. With tissue sequencing, the authors found that neoadjuvant therapy effectively activated the immune system with upregulation of T cell activation-related genes, enhanced HLA expression and a higher phenotype of newly generated TCR clonotypes. Overall, the study was well designed and executed, summarizing the clinical effects of neoadjuvant therapy and providing a more in-depth analysis of clinical phenomena at the genetic level. However, I still have some questions and believe that minor revisions are required.

Response:

We appreciate the positive evaluation from the reviewer, as well as the suggestions that will definitely improve the quality of our manuscript.

1. Early-stage HCC has a relatively favorable prognosis, and more explanation is

needed as to the rationale for choosing early-stage HCC (3 of which were BCLC stage 0) as the study population, why not intermediate or late stage?

Response:

This question is very important to establish the clinical rationale of neoadjuvant therapy for early-stage HCC. In the “Discussion” section, we have discussed three reasons why the effective neoadjuvant therapies for early-stage HCC is worth exploring:

1. HCC is highly invasive, 1-year post-operative recurrent rate of BCLC 0-A stage HCC is over one quarter (26.8%), and the 3-year and 5-year DFS were only 54.6% and 45.4%, respectively.
2. Patients with early-stage HCC normally have relatively better liver function reserve, good performance status, and greater tolerance to potential immune-related toxicities.

3. In case of being refractory to the treatment, early-stage HCC has lower risk of progressing to unresectable disease, which is a major concern for neoadjuvant therapies.

In the revised manuscript, we gave more explanations to this issue:

4. The nature of this study was clearly defined as “neoadjuvant” therapy of HCC, which means the tumor of the participants must be clearly and definitely “resectable”; that is, the recommended first treatment option should be surgical resection. In BCLC staging system, only for stage 0-A tumor, the recommended 1st treatment is resection/(ablation/transplantation); while for tumors of stage B or beyond, the most 1st option is not resection, but local or systemic therapy. In these scenario, the nature of the treatment would rather be “conversion” or “down-staging”, instead of “neoadjuvant”.
5. SBRT has potential immune modulating function, so when radiotherapy is combined with ICIs, SBRT is the most frequently used technique¹². But SBRT is technically not suitable for large or multiple tumors.

Taking together, we believe the neoadjuvant SBRT+ICI in early-stage HCC is worth further studying.

2. One patient received radiofrequency ablation, other 19 patients achieved curative R0 resection. When calculating the ORR and DCR, why not use the 20 people enrolled in the group? In Figure 4, the results for 20 patients are shown?

Response:

Sorry for the confusion that caused. Based on the predefined analysis set in the statistical plan of the trial, “all participants who complete at least one dose of tislelizumab and one fractions of SBRT will be included in the safety analysis (SAS). All participants in SAS who complete curative HCC resection will be included in the efficacy analysis (EAS)”, we analyzed tumor response (CR, PR, ORR, DCR, pathological response, etc.), survival (OS, DFS, etc.) in EAS, in which patient No.1 who received RF ablation was not included. To avoid confusion, we have removed patient No.1 in figure 4, and have showed the results of 19 patients to keep consistent with efficacy results.

3. The study included 4 cases of recurrent tumors; does this have an impact on the

results? Two of these were recurrent within two years of initial resection. It is generally accepted that early recurrence (within 2 years) is associated with the primary tumor and in some way implies higher aggressiveness.

Response:

Thanks for the reviewer's suggestion, it's a great idea to study whether recurrent or highly invasive HCC responds differently to the neoadjuvant therapy.

At the data cutoff (Dec 1st, 2023) for the current analysis, all 4 recruitments with recurrent HCC remain in the status of DFS. As we have mentioned in the "Adjuvant therapy" section in the "Results", "patient no.2 discontinued (adjuvant therapy) after 4 cycle due to cerebral infarction"; patient no.3 had completed all adjuvant PD-1 therapy. These two patients were among the 4 cases with recurrent tumors, and other 2 cases are still receiving regular adjuvant PD-1 therapy according to the protocol.

So, in general, the follow-up time of this study is still very short; as we stated in the manuscript, the median follow-up was only 4.0 months since the resection (range, 2.2-18.8), and disease recurrence developed in only 1 out of 19 patients (5.3%, pts no.

8). We will keep a close eye on these 4 patients, and see if we can find some different trend in the survival data after long-term follow-up.

4. Changes in immune-activated T cells before and after treatment were clearly observed in CR patient number 4 and 8 (Fig. 8), but not much in number 12; does this suggest that the initial immune status prior to treatment also has an impact on the treatment outcome?

Response:

Indeed, patients with more active immune status prior to treatment usually showed a better treatment outcome in previous reports, it seems understandable that baseline immune cells infiltration in the TME would play an important role not only in tumor control prior to treatment, but also in the response to immune checkpoint blockade.

An association between the density of pre-existing CD⁸⁺ T cells located at the invasive tumor margin and the response to anti-PD-1 treatment (pembrolizumab) has been especially demonstrated in patients with metastatic melanoma¹³. In HCC, it has been shown that the baseline density of TILs and treatment response are correlated.

The subgroup analysis of the Checkmate 040 study indicated that CR and PR patients

exhibited a higher CD3⁺ TILs frequency than those with SD. Furthermore, an increase in CD3 and CD8 TILs represented a trend toward enhanced OS, although not statistically significant¹⁴.

In our study, we found immune activation from treatment also play an important role in treatment outcome, and this phenomenon was not limited in those patients with active immune status pre-treatment. The fold change of immune activation from the treatment seems to be less significant in pt 12 than in pt 4 and 8, probably due to the high baseline level of immune activation in pt12.

We hope further similar but large-scale trials with enough CR patients and relevant biomarker study can answer this question with confidence.

5. In lines 528-554 of the discussion section, the authors mention disease progression during neoadjuvant therapy as a concern and use 2 studies as examples. In lines 552-554, the authors claim that “the combination of a local-regional therapy to ICIs can significantly improve the local control rate and avoid the cancellation of curative resection due to tumor progression after neoadjuvant therapy”. Although no tumor progression was observed in this study, this conclusion seems inappropriate, and the

authors then analyze the limitations of the study as well.

Response:

We revised the expression and tone of this part:

“From a more general point of view, our preliminary results could bring more follow-up large-scale trials to validate, whether the combination of a local-regional therapy to ICIs can significantly improve the local control rate, thus reduce the risk of cancellation of curative resection due to tumor progression after neoadjuvant therapy.”

6. The author repeats many of the same items in the discussion section as in the results section, which could be considered appropriately refined.

Response:

In the “Discussion” section, we removed the content that discussing the cohort of “203 HCC patients treated with upfront curative resection” to the supplementary document. We also deleted simple data results, which had been presented in the

“Results” section, and focused on discussing the indications, significances and conclusions derived from these data.

7. Is postoperative adjuvant PD-1 monoclonal antibody necessary for patients with pCR?

Response:

This is a great question with clinical significance. Actually, it is one of the two most frequently discussed questions in the field of neoadjuvant/conversion therapy, the other one is: for patients achieving radiographic CR after the neoadjuvant/conversion therapy, is the surgery still 100% necessary? is “watch and wait” policy also a possible choice? To answer these two questions, solid and reliable RCT data is definitely needed, but still missing.

In adjuvant therapy, chemotherapeutics (and TKIs as well) aim to directly kill residual cancer cells to improve the survival, so for patients with pCR, theoretically, the rationale of further adjuvant therapy is not as solid as in patients who fail to achieve pCR. But inside the wards, the clinical practice is always more complicated. For example, in a PSM analysis with 741 rectal cancer patients in each group, the results

indicated that adjuvant chemotherapy was still associated with improved OS in pCR patients after neoadjuvant chemoradiotherapy¹⁵.

Instead of killing cancer cells directly, neoadjuvant/adjuvant ICIs aim to boost anti-tumor immunity, thus achieving the survival benefit. It is hypothesized that due to the presence of tumor antigens presented before resection, neoadjuvant ICI therapy may be more effective compared to the adjuvant therapy¹⁶. On the basis of neoadjuvant ICIs, whether adjuvant ICI can help to better maintain or further enhance the anti-tumor immunity, the trials of “neoadjuvant ICI only vs. neoadjuvant ICI + adjuvant ICI” are needed, and we don’t have an answer now.

So far, in the literatures, studies of neoadjuvant ICIs normally compared “neoadjuvant ICI + adjuvant ICI vs. adjuvant ICI only”, and pilot results in melanoma showed a better survival in the former “sandwiched” modality^{17, 18, 19}. So, in our trial, we followed this pattern; all patients were designed to receive adjuvant tislelizumab unless clinically unsuitable or contraindications arise.

Our sample size was limited, and we have only 2 pCR. In large-scale trials, if the number of pCR patients after neoadjuvant ICI reaches a certain level, a comparison to

explore the survival significance of adjuvant ICI will be attractive. Or, we can consider whether a meta-analysis to explore this interesting question is possible now.

8. Further description of MHC, HLA, and TCR clonotypes and relationship to immunization and treatment is recommended rather than simply describing the results. AND 9. The activation of immune cells in tumor tissues before and after treatment, and the regulation of related genes is one of the features of this study, which deserves a more in-depth analysis in the discussion section.

Response:

To address these two questions according to the reviewer's requirement, we have basically re-written the later paragraphs of the "Discussion" section.

10. There is a formatting problem with the Trial design in Figure 1 that prevents the entire process from being seen.

Response:

We have re-formatted the figure 1.

Reviewer #4 (Remarks to the Author): with expertise in HCC, therapy

This is a scientifically sound and well-written paper.

Response:

We appreciate the reviewer's positive evaluation to our work.

Please elaborate more on the design to assess and account for adverse events (rolling
6, 3+3?)

Response:

Thanks for the question. For the reviewer' advice to elaborate more on design to
assess and account for adverse events, we revised in 6th paragraph of the "Procedures"
part of the "Methods" section, the details are: "Treatment-related adverse events
(TRAEs) included those events considered by the investigator to be related to study
treatment or with missing assessment of the causal relationship. The incidence of
TRAEs was reported as the number (and percentage). A patient was counted only
once by the highest severity grade."

About the “rolling 6, 3+3”, I suppose the reviewer’s point is, why this Phase Ib study didn’t follow the classic phase I study 3+3 design to explore the study treatment dose from safety perspective. Our consideration was, with the solid efficacy and safety evidence demonstrated from the pivotal study of “RATIONALE 208”²⁰, tislelizumab single-agent had been approved for the 2nd line treatment of advanced HCC in China. Therefore, the tislelizumab in this study follow the package insert dose, which was 200 mg administered on Day 1 of each 21-day cycle, once every 3 weeks.

Please reference the Himalaya trial and the Imbrave150 trials in the introduction and elaborate more on the rationale for using tislelizumab in your trial

Response:

The results from Himalaya trial²¹ and IMbrave 150 trial²² firmly established the pivotal roles of ICIs in the systemic treatment of HCC. We referenced them in the revised manuscript.

We added the following text to the last paragraph of “Introduction” to explain the rational for using tislelizumab: “Tislelizumab is an anti-PD-1 antibody, in the phase

III randomized RATIONALE-301 trial, it demonstrated noninferior overall survival (OS) benefit to sorafenib²³, and has been approved in China as both the first-line and the second-line treatment for patients with unresectable or metastatic HCC.”

Please reference other ongoing and published studies of neoadjuvant RT+/-IO for HCC (work by Ted Hong for example)

Response:

In the first paragraph of the “Discussion” section, we cited other ongoing trials of neoadjuvant radiotherapy (either mono- or combined with ICIs) in HCC.

Dr. Theodore Hong and colleagues have carried out serials of pioneering work in the field of neoadjuvant radiotherapy for pancreatic cancer²⁴, gastric and gastroesophageal cancer²⁵, rectal cancer²⁶, etc. We expect his results in HCC, too.

Reference

1. Marron TU, *et al.* Neoadjuvant cemiplimab for resectable hepatocellular carcinoma: a single-arm, open-label, phase 2 trial. *The Lancet Gastroenterology & Hepatology* **7**, 219-229 (2022).
2. Kaseb AO, *et al.* Perioperative nivolumab monotherapy versus nivolumab plus ipilimumab in resectable hepatocellular carcinoma: a randomised, open-label, phase 2 trial. *Lancet Gastroenterol Hepatol* **7**, 208-218 (2022).
3. Zhang B, *et al.* Real-world practice of conversion surgery for unresectable hepatocellular carcinoma - a single center data of 26 consecutive patients. *BMC Cancer* **23**, 465 (2023).
4. Zhao L, Zhao H. Conversion surgery for hepatocellular carcinoma in the new era of targeted and immune checkpoint inhibitor therapies. *Hepatobiliary Surg Nutr* **9**, 809-811 (2020).
5. Ramos-Casals M, *et al.* Immune-related adverse events of checkpoint inhibitors. *Nat Rev Dis Primers* **6**, 38 (2020).
6. Salem JE, *et al.* Cardiovascular toxicities associated with immune checkpoint inhibitors: an observational, retrospective, pharmacovigilance study. *Lancet Oncol* **19**, 1579-1589 (2018).
7. Shapiro J, *et al.* Neoadjuvant chemoradiotherapy plus surgery versus surgery alone for oesophageal or junctional cancer (CROSS): long-term results of a randomised controlled trial. *Lancet Oncol* **16**, 1090-1098 (2015).
8. Pinato DJ, *et al.* PRIME-HCC: phase Ib study of neoadjuvant ipilimumab and nivolumab prior to liver resection for hepatocellular carcinoma. *BMC Cancer* **21**, 301 (2021).

9. Darragh LB, *et al.* A phase I/Ib trial and biological correlate analysis of neoadjuvant SBRT with single-dose durvalumab in HPV-unrelated locally advanced HNSCC. *Nat Cancer* **3**, 1300-1317 (2022).
10. Chen G, *et al.* Effects of neoadjuvant stereotactic body radiotherapy plus adebrelimab and chemotherapy for triple-negative breast cancer: A pilot study. *Elife* **12**, (2023).
11. Li M, *et al.* A Phase 1b Clinical Trial of Neoadjuvant Radio-immunotherapy for Esophageal Squamous Cell Cancer. *Int J Radiat Oncol Biol Phys*, (2024).
12. Palermo B, *et al.* Stereotactic Ablative Radiation Therapy in 3 Fractions Induces a Favorable Systemic Immune Cell Profiling in Prostate Cancer Patients. *Oncoimmunology* **12**, 2174721 (2023).
13. Tumeh PC, *et al.* PD-1 blockade induces responses by inhibiting adaptive immune resistance. *Nature* **515**, 568-571 (2014).
14. Sangro B, *et al.* Association of inflammatory biomarkers with clinical outcomes in nivolumab-treated patients with advanced hepatocellular carcinoma. *J Hepatol* **73**, 1460-1469 (2020).
15. Polanco PM, Mokdad AA, Zhu H, Choti MA, Huerta S. Association of Adjuvant Chemotherapy With Overall Survival in Patients With Rectal Cancer and Pathologic Complete Response Following Neoadjuvant Chemotherapy and Resection. *JAMA Oncol* **4**, 938-943 (2018).
16. Versluis JM, Long GV, Blank CU. Learning from clinical trials of neoadjuvant checkpoint blockade. *Nat Med* **26**, 475-484 (2020).

17. Rozeman EA, *et al.* Survival and biomarker analyses from the OpACIN-neo and OpACIN neoadjuvant immunotherapy trials in stage III melanoma. *Nat Med* **27**, 256-263 (2021).
18. Blank CU, *et al.* Neoadjuvant versus adjuvant ipilimumab plus nivolumab in macroscopic stage III melanoma. *Nat Med* **24**, 1655-1661 (2018).
19. Song Y, *et al.* Neoadjuvant Versus Adjuvant Immune Checkpoint Blockade in the Treatment of Clinical Stage III Melanoma. *Ann Surg Oncol* **27**, 2915-2926 (2020).
20. Ren Z, *et al.* Tislelizumab in Patients with Previously Treated Advanced Hepatocellular Carcinoma (RATIONALE-208): A Multicenter, Non-Randomized, Open-Label, Phase 2 Trial. *Liver Cancer* **12**, 72-84 (2023).
21. Abou-Alfa GK, *et al.* Tremelimumab plus durvalumab in unresectable hepatocellular carcinoma. *NEJM Evidence* **1**, EVIDoA2100070 (2022).
22. Finn RS, *et al.* Atezolizumab plus Bevacizumab in Unresectable Hepatocellular Carcinoma. *N Engl J Med* **382**, 1894-1905 (2020).
23. Qin S, *et al.* Tislelizumab vs Sorafenib as First-Line Treatment for Unresectable Hepatocellular Carcinoma: A Phase 3 Randomized Clinical Trial. *JAMA Oncol* **9**, 1651-1659 (2023).
24. Murphy JE, *et al.* Total Neoadjuvant Therapy With FOLFIRINOX in Combination With Losartan Followed by Chemoradiotherapy for Locally Advanced Pancreatic Cancer: A Phase 2 Clinical Trial. *JAMA Oncol* **5**, 1020-1027 (2019).
25. Kim DW, *et al.* Neoadjuvant versus Postoperative Chemoradiotherapy is Associated with Improved Survival for Patients with Resectable Gastric and Gastroesophageal Cancer. *Ann Surg Oncol* **29**, 242-252 (2022).

26. Hall WA, *et al.* Prospective Correlation of Magnetic Resonance Tumor Regression Grade With Pathologic Outcomes in Total Neoadjuvant Therapy for Rectal Adenocarcinoma. *J Clin Oncol* **41**, 4643-4651 (2023).

REVIEWERS' COMMENTS

Reviewer #1 (Remarks to the Author):

The authors have satisfactorily addressed all my comments in their response.

Reviewer #3 (Remarks to the Author):

In this study, the authors made a great deal of work to explore new options for early HCC treatment options, which helps to promote the development of precision and personalization of HCC treatment. In addition, the authors were very conscientious and rigorous in revising and optimizing the manuscript, and the overall logic and readability of the manuscript has improved. The authors provided convincing answers to the questions raised by the reviewers. This work has positive significance for the treatment of early HCC.

Reviewer #4 (Remarks to the Author):

Congratulations again on this work!

REVIEWER COMMENTS

Reviewer #1 (Remarks to the Author):The authors have satisfactorily addressed all my comments in their response.

Response:

We are very grateful for your professional advice and help in the process of revising our article, which has helped us improve our article.

Reviewer #3 : (Remarks to the Author):

In this study, the authors made a great deal of work to explore new options for early HCC treatment options, which helps to promote the development of precision and personalization of HCC treatment. In addition, the authors were very conscientious and rigorous in revising and optimizing the manuscript, and the overall logic and readability of the manuscript has improved. The authors provided convincing answers to the questions raised by the reviewers. This work has positive significance for the treatment of early HCC.

Response:

We appreciate the reviewer's suggestions that definitely improved the quality of our manuscript.

Reviewer #4 (Remarks to the Author):

Congratulations again on this work!

Response:

We appreciate the reviewer's positive evaluation to our work and suggestions for revising the article.